# Multiple E3 ligases control tankyrase stability and function

Jerome Perrard [1] & Susan Smith [1] ✉

Tankyrase 1 and 2 are ADP-ribosyltransferases that catalyze formation of polyADP-Ribose (PAR) onto themselves and their binding partners. Tankyrase protein levels are regulated by the PAR-binding E3 ligase RNF146, which promotes K48-linked polyubiquitylation and proteasomal degradation of tankyrase and its partners. We identified a novel interaction between tankyrase and a distinct class of E3 ligases: the RING-UIM (Ubiquitin-Interacting Motif) family. We show that RNF114 and RNF166 bind and stabilize mono-ubiquitylated tankyrase and promote K11-linked diubiquitylation. This action competes with RNF146-mediated degradation, leading to stabilization of tankyrase and its binding partner, Angiomotin, a cancer cell signaling protein. Moreover, we identify multiple PAR-binding E3 ligases that promote ubiquitylation of tankyrase and induce stabilization or degradation. Discovery of K11 ubiquitylation that opposes degradation, along with identification of multiple PAR-binding E3 ligases that ubiquitylate tankyrase, provide insights into mechanisms of tankyrase regulation and may offer additional uses for tankyrase inhibitors in cancer therapy.

Tankyrase 1 and 2 are related multifunctional proteins that act in many cellular pathways and impact human diseases, including cancer[1–4]. Tankyrases have a similar primary structure consisting of a C-terminal catalytic PARP domain, a SAM (sterile alpha module) domain, an ankyrin repeat domain, and an N-terminal HPS (His, Pro, and Ser) domain of unknown function, unique to tankyrase 1[5]. The SAM domain promotes homo- and hetero-oligomerization of tankyrase 1 and 2, which is required for full catalytic activity[6–9]. The ankyrin domain is organized into five ankyrin repeat clusters (ARCs), which serve as a basic unit for recognizing a tankyrase-binding motif (TBM) Rxx(G/P/A)xGxx in its binding partners[10–14]. A distinguishing feature of tankyrases is their ability to interact through the ARCs with a broad range of binding partners[15].

Proteomic and in silico screens have identified hundreds of potential tankyrase-binding proteins (TBPs)[12,16,17]. Over 40 human TBPs have been validated by co-immunoprecipitation; almost all contain an RxxxxG sequence that binds to the ARCs of tankyrase 1 or 2[15]. Binding is independent of catalytic activity. Tankyrases localize throughout the cell. A number of tankyrase partners act to recruit tankyrase to a subcellular locale; examples include TRF1-mediated recruitment to telomeres[2,18], NuMA-mediated recruitment to spindle poles[19,20], and IRAP-mediated recruitment to Glut 4 vesicles[21]. Tankyrase 1 and 2 have the same binding partners and mostly overlapping functions[2,15].

An unanticipated function for tankyrase in protein degradation came from a screen for modulators of the Wnt signaling pathway[22]. The majority of colorectal cancers result from activation of this pathway[23]. Wnt controls the stability of the transcriptional coactivator β-catenin. In the absence of the Wnt signal, a cytoplasmic "β-catenin destruction complex" containing the key scaffolding component Axin promotes degradation of β-catenin. A chemical genetic screen for inhibitors of this pathway identified XAV939, a small molecule inhibitor of tankyrase[22]. Tankyrase was shown to positively modulate this pathway; tankyrase-mediated PARylation of Axin led to its degradation, resulting in β-catenin stabilization[22].

Subsequently, RNF146 was identified as the PAR-directed RING E3 ligase that regulates the degradation of Axin[24,25]. RNF146 interacts with PARylated substrates through its internal WWE domain that binds to iso-ADP-ribose, the internal unit of the PAR polymer[26–28]. RNF146 promotes K48-linked polyubiquitylation and degradation of PARylated tankyrase and PARylated targets, including itself. Many targets

¹Department of Cell Biology, New York University School of Medicine, New York, NY 10016, USA. ✉e-mail: susan.smith@med.nyu.edu

have been identified, including 3BP2 (c-ABL SH3 domain binding protein 2)[29]; BLZF1 (basic leucine zipper factor 1)[25]; PTEN, a critical tumor suppressor[30]; and AMOT (Angiomotin), a cancer cell signaling protein[31]. A whole proteome screen for proteins stabilized in HEK293T cells deleted for tankyrases (TNKS1/2 DKO) identified most of these and many additional proteins[16]. Thus, tankyrase-mediated degradation can impact a range of cellular targets and pathways[3].

Considering the role of tankyrase in diverse pathways, we sought to determine if there were E3 ligases (in addition to RNF146) that could influence the stability of PARylated tankyrase and its partners. Here we describe a novel interaction between tankyrase and a distinct class of E3 ligases: the RING-UIM (Ubiquitin-Interacting Motif) family[32,33]. We show that unlike all other tankyrase-binding proteins, which interact with the ankyrin domain, the RING-UIMs (specifically RNF114 and RNF166) bind the catalytic SAMPARP domain. We show that RNF166 promotes K11-linked diubiquitylation of tankyrase, dependent on tankyrase catalytic activity. This novel tankyrase modification competes with RNF146-mediated K48-linked polyubiquitylation and degradation to promote the stabilization of tankyrase and at least one binding partner, Angiomotin. We additionally identify several PAR-binding E3 ligases that can influence tankyrase levels. Together our work reveals a complex network for regulating levels of PARylated tankyrase and its partners.

## Results
### Tankyrase binds RING-UIM E3 ligases
A previous proteomic screen used an immunoprecipitation/mass spectrometry approach to identify tankyrase binding partners from HEK293T cells overexpressing tankyrase 1 or 2[17]. To distinguish binding partners that were dependent on catalytic activity, cells were treated with the tankyrase inhibitor (TNKSi) XAV939 or DMSO as control. Over 100 significant (known and novel) targets were identified in the XAV939-treated cells that overlapped substantially with the DMSO control group. This was expected, since almost all tankyrase-binding proteins bind the ankyrin domain, independent of tankyrase catalytic PARP activity. However, a small number of proteins were highly enriched in the DMSO-treated control compared to the XAV939-treated cells, indicating a role for catalytic activity. Among them were several E3 ligases: the known PAR-binding E3 ligase RNF146 and three E3 ligases (RNF114, RNF166, and RNF138) that drew our attention because they were all members of the same RING-UIM E3 ligase family[32,33]. The family, which has four members (with the addition of RNF125), is not well characterized. RNF114, 166, and 125 have been reported in association with innate immunity, IFN-signaling pathways, and T cell activation[33], and RNF138 in association with DNA repair[34]. Their primary structure is comprised of an N-terminal RING E3 ligase domain followed by a single C2HC ZnF (zinc finger), two atypical C2H2 ZnFs that resemble the Zn finger-like domains of the plant drought-induced Di19 gene family[35], and a C-terminal ubiquitin interacting motif (UIM) (Fig. 1A). Unlike RNF146, they do not contain a WWE domain or any other known PAR-binding motif. RNF166 contains two potential TBMs and RNF125 a myristylation site (Fig. 1A).

We initially investigated RNF166 since it (unlike the other three) contained potential TBMs. We transfected Flag-epitope-tagged RNF166 (FlagRNF166) and vector or FlagBAP (bacterial alkaline phosphatase) as a negative control into HeLa cells (Fig. 1B). To test the effect of inhibiting tankyrase catalytic activity, the RNF166 transfected cells were treated with tankyrase inhibitor #8 (TNKSi)[36]. Note that TNKSi leads to the stabilization of tankyrase (Fig. 1B, Input lane 4) because it prevents autoPARylation and degradation by RNF146[24,25]. The Flag IP shows that FlagRNF166 (but not BAP or vector) coimmunoprecipitated endogenous TNKS1 (Fig. 1B, IP lane 7). This was reduced by TNKSi treatment (Fig. 1B, IP lane 8), despite increased levels of TNKS1 in TNKSi cells. These data indicate a robust interaction between TNKS1 and RNF166 that is dependent on TNKS1 catalytic activity, consistent

with the identification of RNF166 (specifically in the absence of TNKSi) in the screen described above. To determine if the RNF166 potential TBMs are required for TNKS1 binding, we generated point mutations in the essential Gly of each potential TBM: G19A and G47A, as well as a double mutation 2GA (see Fig. 1A). As shown in Fig. 1C, FlagRNF166 (WT or mutants) robustly coimmunoprecipitated endogenous TNKS1, indicating that TNKS1 binding to RNF166 is independent of the candidate TBMs.

To further analyze the RING-UIM E3 ligase family, we introduced each member into TNKS1/2 DKO HEK293T[16] cells along with TNKS1. As shown in Fig. 1D, RNF114 and RNF166 expression led to a dramatic increase in TNKS1 protein level along with a shift in migration (lanes 2 and 3). RNF125 and 138 had no effect, perhaps due to their low level of expression. RNF146 led to a reduction in TNKS1 (lane 6), consistent with its role in degradation. The Flag IP shows that all four RING-UIMs and RNF146 (but not BAP) coimmunoprecipitated TNKS1, most notably RNF114 (lane 8) and RNF166 (lane 9).

We next performed the same analysis in the presence or absence of TNKSi. As shown in Fig. 1E, the shift in migration of TNKS1 induced by RNF114 and RNF166 in the Input samples was abrogated by TNKSi treatment (Fig. 1E, lanes 4 and 6), as was the robust immunoprecipitation (Fig. 1E, lanes 14 and 16). TNKSi had no effect on the interaction between TNKS1 and RNF125 or 138 (lanes 18 and 20). In the presence of TNKSi, all four members immunoprecipitated low levels of TNKS1 (Fig. 1E, lanes 14, 16, 18, and 20). The lack of effect of RNF125 and 138 on TNKS1 could be due to their low level of expression. However, even when expressed at similar levels as RNF166, RNF125 and RNF138 did not impact TNKS1 (Supplementary Fig. 1A). Thus, all four members bind TNKS1, but only RNF114 and 166 impact TNKS1 level/modification and do so in a manner that requires TNKS1 catalytic activity.

The stimulation of TNKS1 levels by RNF114 and RNF166 was robust and reproducible. Quantification showed that RNF114 or 166 induced a two to three-fold increase in tankyrase protein (Fig. 1F, top panel, lanes 3 and 5 and Fig. 1G). This stimulation was abrogated by treatment with TNKSi (lanes 4 and 6 and Fig. 1G). We observed some increase in TNKS1 in the presence of TNKSi, but the effect was distinct as it was only with RNF166 and only on the unmodified form of TNKS1. Immunoblot analysis with antibody against PAR showed a two-fold increase in PAR-TNKS levels (Fig. 1F, middle panel, lanes 3 and 5 and Fig. 1H) that was reduced in the presence of TNKSi (Fig. 1F middle panel, lanes 4 and 6 and Fig. 1H). Note, the PAR antibody also detects a faster migrating band at approximately 130 kDa that, based on its detection with anti-PARP1 antibody and its abrogation following treatment with the PARP inhibitor olaparib (Supplementary Fig. 1B), corresponds to PAR-PARP1. Together these data show that RNF114 and RNF166 stimulate an increase overall in the level of PARylated tankyrase.

We next compared the interaction of the RING-UIM E3s with wild-type (WT) TNKS1 versus a catalytically dead (CD) mutant (TNKS1.HE/A)[37]. RNF114 and 166 stimulated TNKS1 WT (but not CD) modification (Supplementary Fig. 1C; lanes 2 and 6) and immunoprecipitation (lanes 14 and 18), consistent with the effect of TNKSi. We observed some stimulation of TNKS1 CD in the Input (lane 12), but as described above in Fig. 1F for TNKSi, the effect was only with RNF166 and only on the unmodified form of TNKS1. To determine if the RNF114/166 preference for catalytically active TNKS1 was due to a preference for PARylated protein, we probed for coimmunoprecipitation of PARP1. As shown in Supplementary Fig. 1C, only RNF146, which binds to PARylated PARP1 through its PAR-binding WWE domain, coimmunoprecipitated PARP1 (lanes 17 and 23). Finally, we asked if RNF166 (like RNF146) is PARylated by tankyrase. We introduced BAP, FlagRNF146, or FlagRNF166 into PARP1[−/−]/PARP2[−/−] cells[38] (to ensure that any PARylation detected was due to tankyrase) and performed Flag immunoprecipitation followed by immunoblot with anti-PAR antibody (Supplementary Fig. 1D). We detect PARylated RNF146 (and PARylated TNKS1), but not PARylated RNF166, indicating that RNF166 is not PARylated by TNKS1.

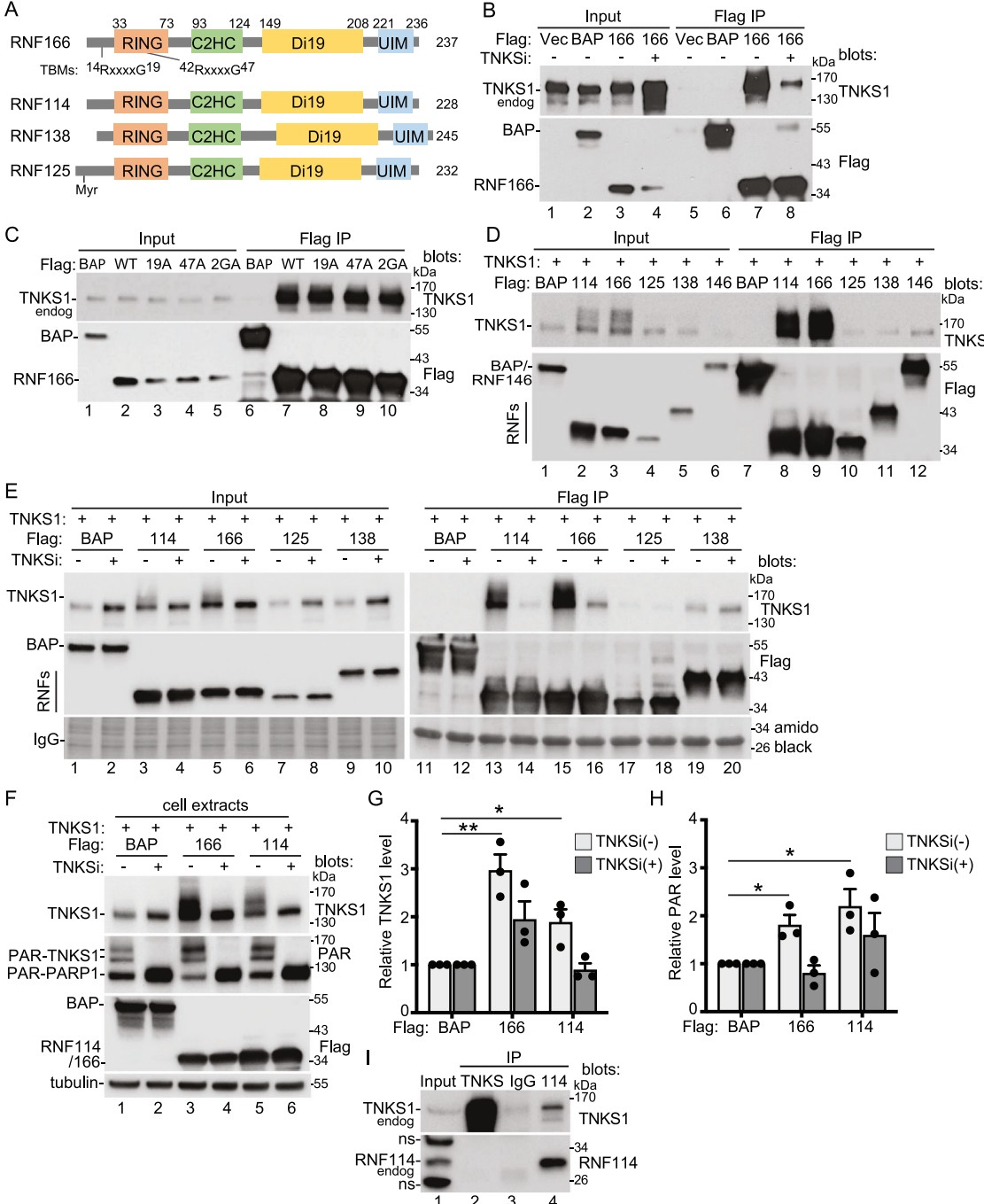

**Fig. 1 | RING-UIM E3 Ligases bind tankyrase. A** Schematic diagram showing the primary structure of the RING-UIM E3 ligase family. The TBMs, [14]RQPPAGPA and [42]RPVAIGSC, are shown as RxxxxG. **B** Immunoblot analysis of SuperHeLa cells transfected with the indicated Flag plasmids, with or without TNKSi, immunoprecipitated with anti-Flag antibody, and probed with the indicated antibodies. At least three independent experiments produced similar results. **C** Immunoblot analysis of U2OS cells transfected with the indicated Flag plasmids, immunoprecipitated with anti-Flag antibody, and probed with the indicated antibodies. Two independent experiments produced similar results. **D** Immunoblot analysis of TNKS1/2 DKO HEK293T cells transfected with TNKS1 and the indicated Flag plasmids, immunoprecipitated with anti-Flag antibody, and probed with the indicated antibodies. At least three independent experiments produced similar results. **E** Immunoblot analysis of TNKS1/2 DKO HEK293T cells transfected with TNKS1 and the indicated Flag plasmids, with or without TNKSi, immunoprecipitated with anti-Flag antibody, and probed with the indicated antibodies or stained with amido black. At least two

independent experiments produced similar results. **F** Immunoblot analysis of TNKS1/2 DKO HEK293T cells transfected with TNKS1 and the indicated Flag plasmids, and probed with the indicated antibodies. At least three independent experiments produced similar results. **G** Graphical presentation of the relative increase in TNKS1 levels induced by RNF166 and 114 relative to tubulin and normalized to the BAP control (−) or (+) TNKSi. Average of three independent experiments ± SEM. BAP vs 166: $p = .004$, BAP vs 114: $p = .029$. $*p \leq 0.05$, $**p \leq 0.01$, Student's unpaired two-sided t-test. **H** Graphical presentation of the relative increase in PAR levels induced by RNF166 and 114 relative to tubulin and normalized to the BAP control (−) or (+) TNKSi. Average of three independent experiments ± SEM. BAP vs 166: $p = .016$, BAP vs 114: $p = .028$. $*p \leq 0.05$, Student's unpaired two-sided t-test. **I** Immunoblot analysis of HEK293T cell extract after immunoprecipitation with anti-TNKS1, anti-IgG or anti-RNF114 antibodies, and probed with the indicated antibodies. ns; nonspecific. At least two independent experiments produced similar results. Source data are provided as a Source Data file.

Lastly, we showed an interaction between endogenous proteins. Immunoprecipitation of endogenous RNF114 with anti-RNF114 antibody coimmunoprecipitated endogenous TNKS1 from HEK293T cells (Fig. 1I, Lane 4). We did not detect endogenous RNF114 in the TNKS1 immunoprecipitation, which is often the case due to the great number of TBPs.

Together our data indicate that all four RING-UIM E3s interact with TNKS1. RNF114 and 166 lead to a reproducible increase in the level of tankyrase protein, an altered migration, and a robust coimmunoprecipitation, dependent on TNKS1 catalytic activity. RNF125 and RNF138 bind TNKS1, but do not affect TNKS1 levels or modification.

## Map the interacting domains between RNF166 and TNKS1

The RING-UIM E3s have four domains. We generated mutations or deletions in each domain of RNF166 to determine which domains are necessary for binding (Fig. 2A). For the N-terminal catalytic RING, we

mutated the essential ZnF Cysteines (C33S; C36S; R**)[39]. For the adjacent C2HC domain, which has been shown to cooperate with the RING domain[40], we mutated the first Cysteine (C98G; C*) of the ZnF. For the Di19, whose function is unknown, we generated separate point mutations in the first cysteine of each C2H2 ZnF (C152R; D1* and C182G; D2*). For the UIM, we deleted the C-terminal 17 amino acids, 221 to 237 (UIMΔ; UΔ). We transfected the mutants and measured coimmunoprecipitation of endogenous tankyrase (Fig. 2B). Two classes of mutants were observed. The RING, UIMΔ, and C2HC mutants retained partial binding (lanes 10, 11, and 12), whereas each of the Di19 mutants showed no binding (lanes 13 and 14). Since some of the RNF166 mutants showed a lower expression than RNF166 WT we repeated the analysis. TNKS1 was cotransfected with various concentrations of DNA for the RNF166 mutants to obtain comparable levels of RNF166 mutant proteins. As shown in Fig. 2C, even when the mutant proteins were expressed at levels similar to or greater than the WT, we observed the

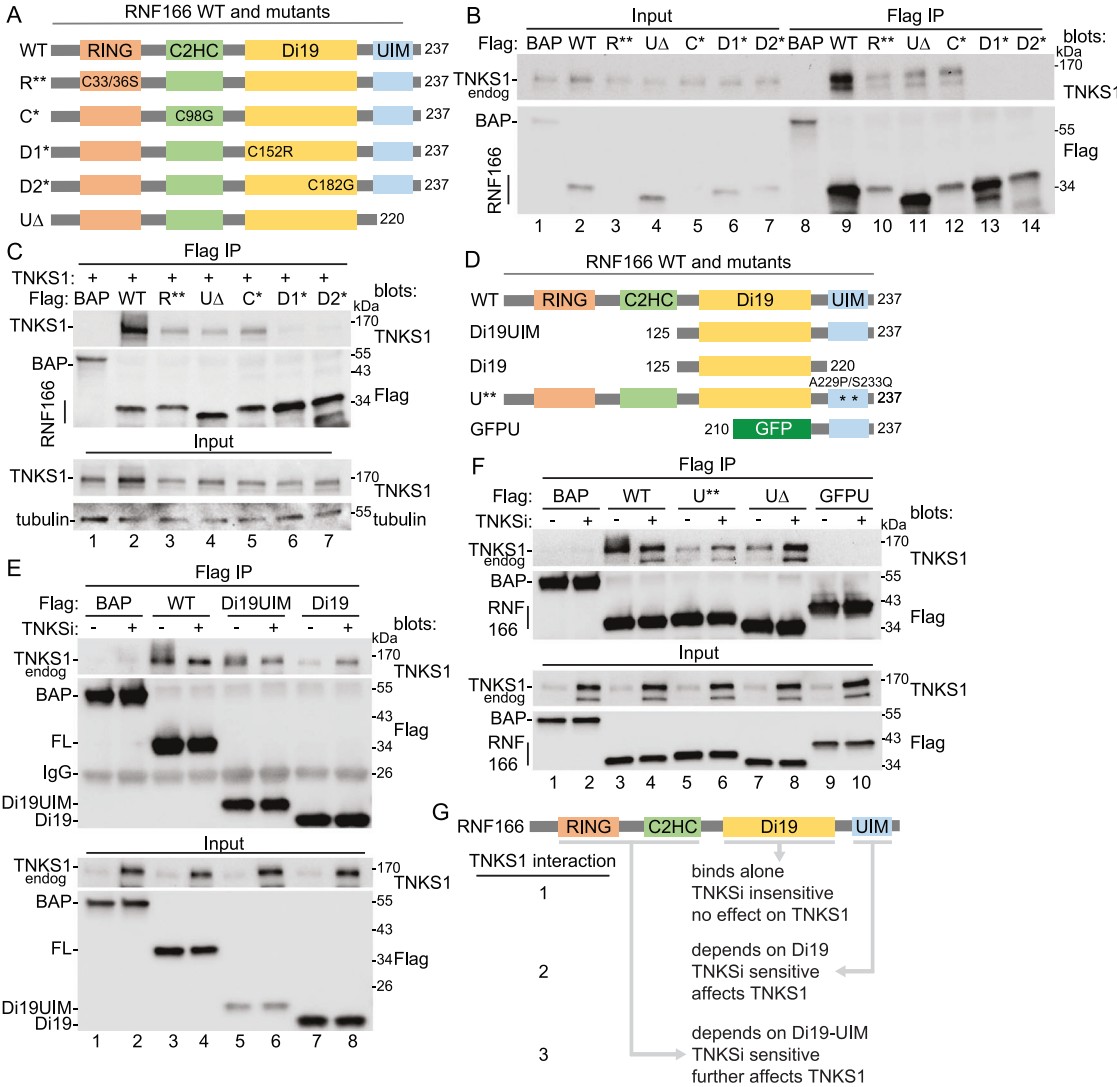

**Fig. 2 | Map the interacting domains of RNF166. A** Schematic diagram showing the RNF166 mutant constructs. **B** Immunoblot analysis of SuperHeLa cells transfected with the indicated Flag plasmids, immunoprecipitated with anti-Flag antibody, and probed with the indicated antibodies. At least two independent experiments produced similar results. **C** Immunoblot analysis of TNKS1/2 DKO HEK293T cells transfected with TNKS1 and the indicated Flag plasmids, immunoprecipitated with anti-Flag antibody, and probed with the indicated antibodies. At least two independent experiments produced similar results. **D** Schematic diagram showing the RNF166 mutant constructs. **E** Immunoblot analysis of HEK293T cells

transfected with the indicated Flag plasmids, immunoprecipitated with anti-Flag antibody, with or without TNKSi, and probed with the indicated antibodies. At least two independent experiments produced similar results. **F** Immunoblot analysis of HEK293T cells transfected with the indicated Flag plasmids, immunoprecipitated with anti-Flag antibody, with or without TNKSi, and probed with the indicated antibodies. At least two independent experiments produced similar results. **G** Schematic diagram showing three levels of interaction between TNKS1 and RNF166. Source data are provided as a Source Data file.

same two classes of binding mutants; partial binding for the RING, UIMΔ, and C2HC mutants (lanes 3, 4, and 5) and no binding for the Di19 mutants (lanes 6 and 7). The Di19 domain appears to be essential for binding, while the other three domains contribute to a lesser extent.

To determine the domains that are sufficient for binding to TNKS1, we generated constructs expressing the Di19 domain alone or with the UIM (Fig. 2D) and measured their ability to coimmunoprecipitate endogenous tankyrase in the presence and absence of TNKSi (Fig. 2E). Note that inhibitor prevents PAR-dependent degradation by RNF146 resulting in increased levels of TNKS1 (see Input). In the presence of TNKSi, Di19 (lane 8), Di19UIM (lane 6), and full length (FL) (lane 4) RNF166 coimmunoprecipitated similar levels of tankyrase (Fig. 2E, top panel). This indicates that the Di19 domain alone is sufficient to bind tankyrase and further, when tankyrase catalytic activity is inhibited, the three constructs bind in a similar way. However, the constructs behave differently in the absence of TNKSi when tankyrase is catalytically active. Here we observed minimal binding with Di19 (lane 7). Addition of the UIM to the Di19 (Di19UIM) led to an increase in TNKS1 binding and a slight shift in migration (lane 5) and the addition of the RING-C2HC to the Di19UIM (to generate the full-length protein) led to even greater binding and a greater shift in migration (lane 3). The increased binding is particularly striking considering the lower level of TNKS1 (in the Input) in the absence of TNKSi.

The results above suggests that the UIM domain stimulates binding to TNKS1 and induces a change in migration of TNKS1. To determine if ubiquitin binding is required for the increase in TNKS1 protein binding and modification that we observe with RNF166 FL, we mutated two of the conserved amino acids in the UIM that are required for ubiquitin-binding (A229P/S233Q; U**) (based on the RNF125 UIM mutant)[32] (Fig. 2D) and compared it side by side with WT RNF166. As shown in Fig. 2F, top panel, WT RNF166 led to a stimulation in TNKS1 binding and a shift in migration (lane 3) that was blocked by TNKSi (lane 4). Mutation of the ubiquitin-binding domain (U**) abrogated the ability of RNF166 to stimulate binding and modification of TNKS1 (lane 5), but still allowed minimal binding in the presence of TNKSi (lane 6), similar to deletion of the UIM (UΔ) (lanes 7 and 8). Together these data indicate a critical role for the UIM in binding and induction of TNKS1 modification. The UIM alone (fused to GFP; GFPU) does not bind TNKS1 in the presence or absence of TNKSi (lanes 9 and 10).

Together these data indicate three levels of interaction between RNF166 and TNKS1 (Fig. 2G). One, the Di19 domain is necessary and sufficient for TNKS1 binding. However, Di19 does not recapitulate RNF166 WT TNKS binding. Unlike RNF166 WT, Di19 binding is insensitive to TNKSi and has no effect on TNKS1 level/modification. Two, the UIM domain binds, but it cannot bind on its own without the Di19 domain. The UIM stimulates TNKS1 level/modification and is sensitive to TNKSi. Three, the RING-C2H2 further stimulates TNKS1 levels and modification and (due to the UIM) is sensitive to TNKSi.

Regarding the Di19 domain, a recent study demonstrated that it binds peptides that specifically contain a mono-ADP-ribose (MAR) modification[41]. This raised the possibility that the Di19 domain might bind a MARylated site on tankyrase. While tankyrase has been shown to be PARylated, its MARylation status has not been determined. We thus performed immunoblot analysis using the high-affinity antibody for MAR used in the study described above[41]. As shown in Supplementary Fig. 2A, TNKS1 is detected by anti-MAR antibody, indicating that TNKS1 is MARylated.

## Determine the interacting domain in tankyrase

We next mapped the domain in tankyrase that is required for RNF166 binding. To date over 40 TBPs have been identified and almost all bind to the tankyrase ARCs using TBMs[15]. As described above, RNF166 has two potential TBMs, but they are not required for binding to TNKS1 and moreover, the other three RING-UIMs, which all bind TNKS1, do

not have candidate TBMs. We thus considered that they might not bind to the ankyrin domain. To address this, we generated constructs to split TNKS1 into two Flag-epitope-tagged domains: HPSAnkyrin (A) and SAMPARP (S) (Fig. 3A), and transfected them into TNKS1/2 DKO cells along with Myc-tagged RNF166 or TRF1, the original and canonical tankyrase binding partner containing an ankryin repeat-binding TBM[5]. As shown in Fig. 3A (lanes 11 and 12), TRF1 binds to the HPS Ankyrin (A), but not to the SAMPARP (S) domain. In contrast, RNF166 binds to the SAMPARP (S), but not to the HPSAnkyrin (A) domain (lanes 9 and 10). This is the first example of a tankyrase-binding protein (other than tankyrase itself) that binds to the SAMPARP and not to the ankyrin domain of tankyrase.

Next, to investigate the catalytic-dependent aspect of RNF166 binding to TNKS1, we compared RNF166 binding side by side with TRF1 binding to full-length endogenous tankyrase in the presence or absence of TNKSi. Flag-tagged BAP, RNF166, Di19UIM, or TRF1, was transfected into HEK293T cells with or without TNKSi, and lysates were immunoprecipitated with anti-Flag antibody (Fig. 3B). TRF1 coimmunoprecipitated TNKS1 (lanes 7 and 8). Binding was greatly increased with TNKSi (lane 8), coincident with the increased level of TNKS1 in the Input. RNF166 and Di19UIM also coimmunoprecipitated TNKS1, but in contrast to TRF1, binding was greatly increased without TNKSi (lanes 3 and 5), which was particularly striking considering the reduced TNKS1 levels in the Input. Thus, RNF166 (and Di19UIM) exhibits an atypical pattern of binding to TNKS1.

Lastly, we asked if the SAMPARP domain of tankyrase was sufficient for interaction with the Di19UIM domain of RNF166. We generated a Myc-tagged SAMPARP construct and cotransfected it into TNKS1/2 DKO cells along with Flag-tagged RNF166 FL or Di19UIM, treated with or without TNKSi, and immunoprecipitated the lysates with anti-Flag antibody. As shown in Fig. 3C, FlagRNF166 (FL or Di19UIM) coimmunoprecipitated MycSAMPARP. Binding was robust in the absence of TNKSi (lanes 9 and 11) and reduced in the presence of TNKSi (lanes 10 and 12). Thus, we can recapitulate the binding and the TNKSi effect observed with the full-length proteins using the SAMPARP domain of tankyrase and the Di19UIM domain of RNF166. Interestingly, we found that the SAMPARP domain (like full length TNKS1) is MARylated (Supplementary Fig. 2A), consistent with the possibility that Di19 could bind tankyrase through MARylation.

## Analyze the RNF166-induced modification on tankyrase

To determine if the RNF166-induced shift in TNKS1 mobility was due to ubiquitylation, we transfected HA-ubiquitin (HA-Ub) along with FlagRNF166 and TNKS1 into TNKS1/2 DKO cells. Cell extracts were immunoprecipitated with anti-HA antibody and blotted for TNKS1 and FlagRNF166. As shown in Fig. 4A, RNF166 induced ubiquitylation of TNKS1 (lane 7). The modification was blocked by TNKSi (lane 8), consistent with the data in Fig. 1 showing that RNF166 increased TNKS1 level/modification, dependent on TNKS1 catalytic activity. Here, we detect two modified forms, which could represent mono and diubiquitylated TNKS. Note that we also detect some unmodified TNKS1, which could be due to heteropolymerization of unmodified with modified tankyrase.

We next sought to recapitulate the ubiquitylation using the smaller interacting domains defined above in Figs. 2 and 3 for each protein: Di19UIM for RNF166 and SAMPARP for TNKS1.

We cotransfected HA-Ub, FlagDi19UIM, and MycSAMPARP into TNKS1/2 DKO cells and immunoprecipitated the cell lysates with anti-HA antibody. As shown in Fig. 4B (lane 7), Di19UIM promotes ubiquitylation of SAMPARP. We observe a single form at a molecular weight consistent with monoubiquitylation. Monoubiquitylation of amino acids on a protein target occurs through the C-terminus of ubiquitin. Subsequent linkages occur though one of seven lysines in the ubiquitin protein itself. To confirm that the observed modification was monoUb, we performed the assay with HA-Ub K0 (in which all seven lysines are

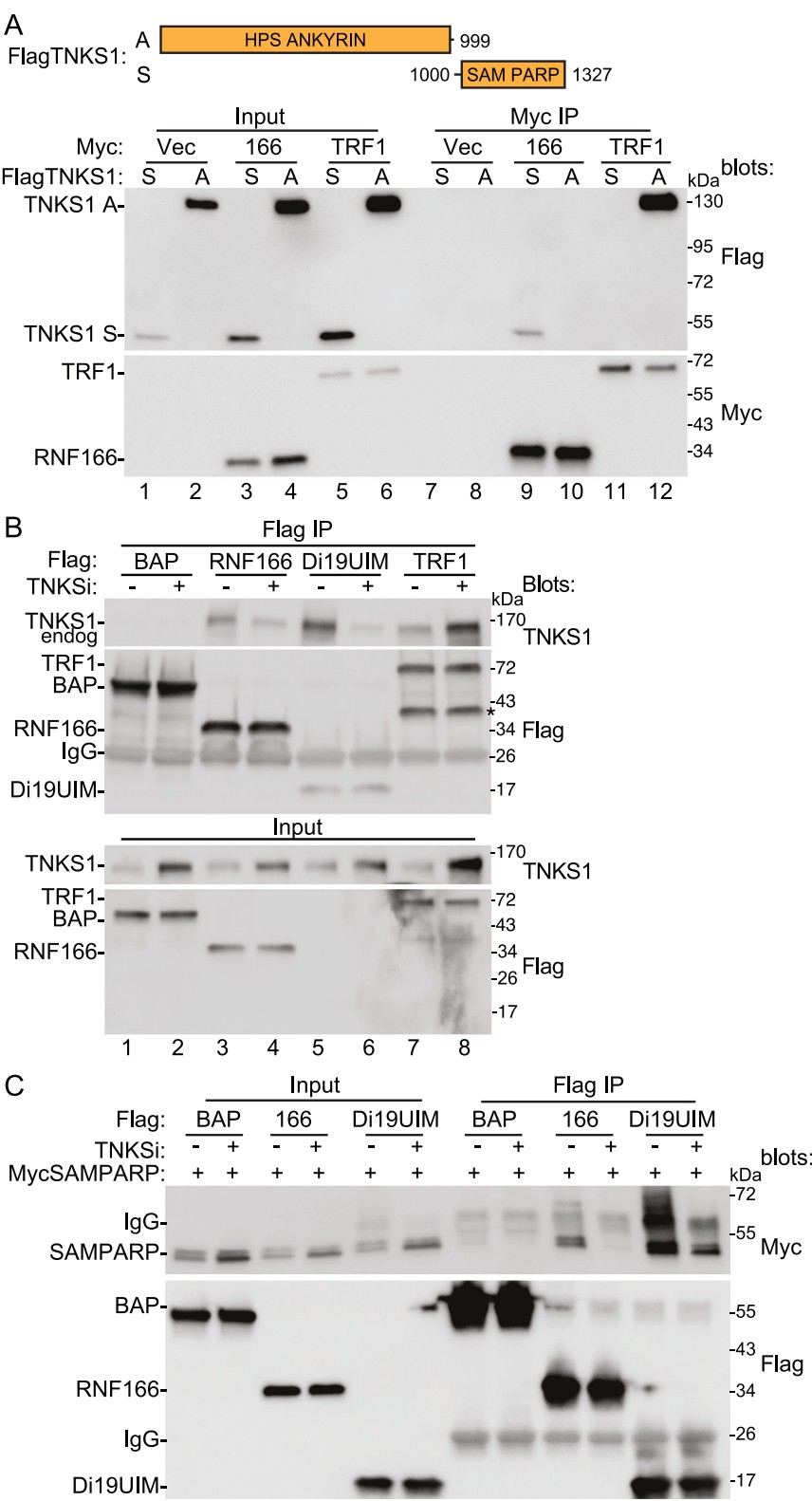

**Fig. 3 | Map the interacting domains of tankyrase. A** Top: schematic representation of tankyrase constructs FlagTNKS1 A and S. Bottom: immunoblot analysis of TNKS1/2 DKO HEK293T cells transfected with the indicated Flag and Myc plasmids, immunoprecipitated with anti-Myc antibody, and probed with the indicated antibodies. At least two independent experiments produced similar results. **B** Immunoblot analysis of HEK293T cells transfected with the indicated Flag plasmids, with or without TNKSi, immunoprecipitated with anti-Flag antibody, and probed with the indicated antibodies. (*) indicates a breakdown product of TRF1. At least three independent experiments produced similar results. **C** Immunoblot analysis of TNKS1/2 DKO HEK293T cells transfected with the indicated Flag and Myc plasmids, immunoprecipitated with anti-Flag antibody, and probed with the indicated antibodies. At least two independent experiments produced similar results. Source data are provided as a Source Data file.

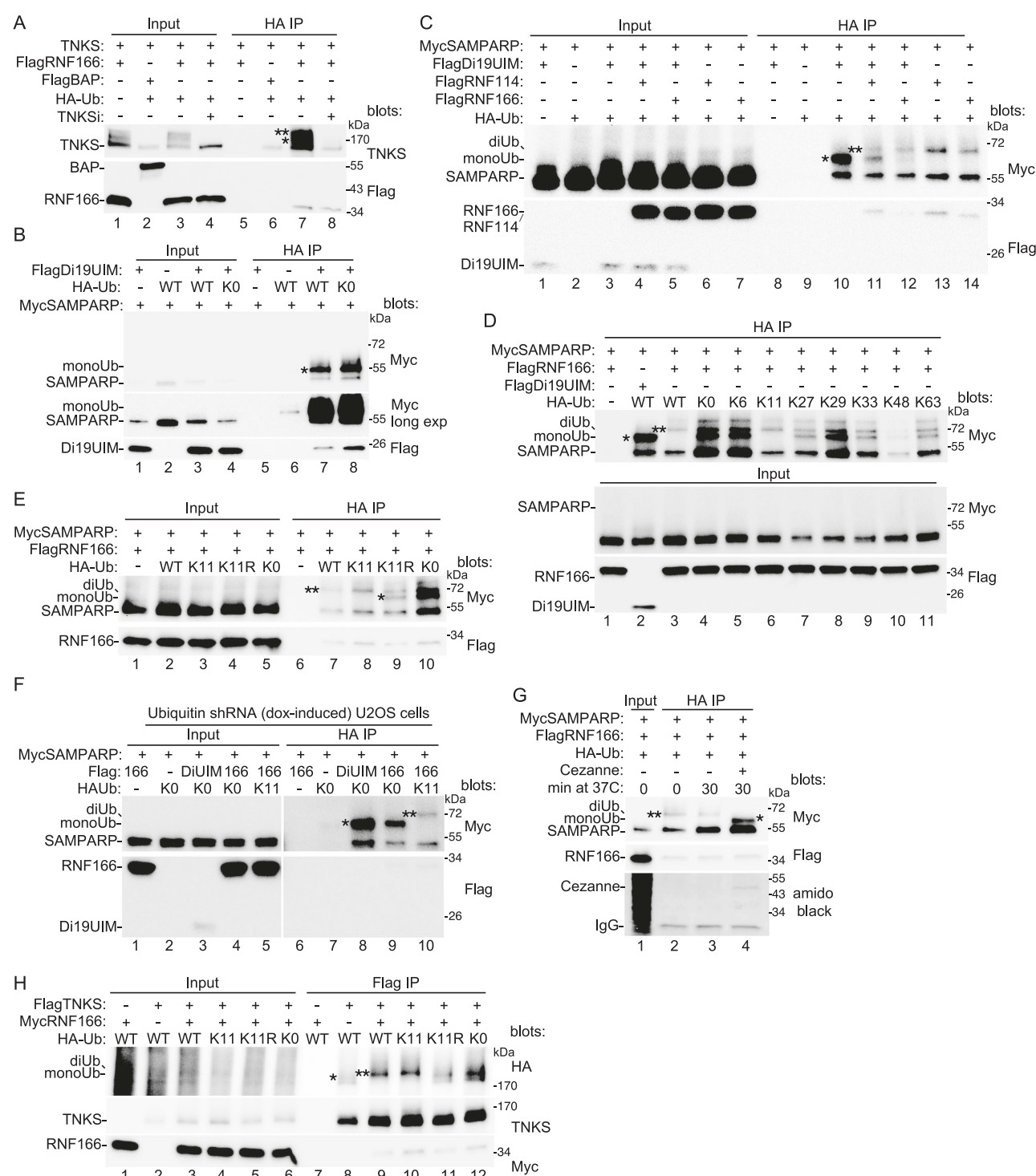

mutated) and observed the same band as with WT Ub, indicating a monoubiquitylation event (Fig. 4B, lane 8). Thus, the Di19UIM domain of RNF166 promotes monoubiquitylation of the SAMPARP domain of tankyrase. Notably, the Di19UIM fragment lacks the catalytic N-terminal E3 ligase RING domain of RNF166 and is therefore unlikely to catalyze the monoubiquitylation event directly, but rather, could instead (through its UIM) bind/stabilize a preexisting mono-UbSAMPARP. Indeed, upon longer exposure we detect mono-UbSAMPARP even in the absence of Di19-UIM (Fig. 4B, lane 6, long exposure).

We next sought to determine the role of the N-terminal RING-C2HC domain. TNKS1/2 DKO cells were cotransfected with HA-Ub,

FlagDi19UIM, and MycSAMPARP (as above in Fig. 4B), additionally, we introduced full-length RNF114 or 166. Ubiquitylated proteins were immunoprecipitated with anti-HA antibody. With the Di19UIM only, we observed monoUbSAMPARP (Fig. 4C, lane 10). Upon addition of RNF114 or RNF166, we observe (in addition to monoUbSAMPARP) a slower migrating form whose migration is consistent with di-UbSAMPARP (lanes 11 and 12, respectively), although western blot-ting does not permit a precise determination of size. If we leave out the Di19UIM and use the full-length RNF114 or RNF166 alone, we detect only di-UbSAMPARP (lanes 13 and 14, respectively). These data indicate that the RING-C2HC domain of RNF114 or 166 promotes diubiquityla-tion of SAMPARP.

**Fig. 4 | Characterize the modification on tankyrase. A** Immunoblot analysis of TNKS1/2 DKO HEK293T cells transfected with TNKS1, the indicated Flag plasmids, and HA-Ub, with or without TNKSi, immunoprecipitated with anti-HA antibody, and probed with the indicated antibodies. At least three independent experiments produced similar results. **B** Immunoblot analysis of TNKS1/2 DKO HEK293T cells transfected with the indicated Flag, Myc, and HA-Ub plasmids, immunoprecipitated with anti-HA antibody, and probed with the indicated antibodies. At least two independent experiments produced similar results. **C** Immunoblot analysis of TNKS1/2 DKO HEK293T cells transfected with the indicated Flag, Myc, and HA-Ub plasmids, immunoprecipitated with anti-HA antibody, and probed with the indicated antibodies. At least two independent experiments produced similar results. **D** Immunoblot analysis of TNKS1/2 DKO HEK293T cells transfected with the indicated Flag, Myc, and HA-Ub plasmids, immunoprecipitated with anti-HA antibody, and probed with the indicated antibodies. At least two independent experiments produced similar results. **E** Immunoblot analysis of TNKS1/2 DKO HEK293T cells transfected with the indicated Flag, Myc, and HA-Ub plasmids, immunoprecipitated

with anti-HA antibody, and probed with the indicated antibodies. At least two independent experiments produced similar results. **F** Immunoblot analysis of Ubiquitin shRNA U2OS cells treated with dox for 48 hr and transfected with the indicated Flag, Myc, and HA-Ub plasmids for 24 h prior to harvest. Cell lysates were immunoprecipitated with anti-HA antibody, and probed with the indicated antibodies. Two independent experiments produced similar results. **G** Immunoblot analysis of TNKS1/2 DKO HEK293T cells transfected with the indicated Flag, Myc, and HA-Ub plasmids, immunoprecipitated with anti-HA antibody, and probed with the indicated antibodies or stained with amido black. Samples were incubated for 0 or 30 min at 37 °C with or without Cezanne prior to loading on the gel. At least three independent experiments produced similar results. **H** Immunoblot analysis of TNKS1/2 DKO HEK293T cells transfected with the indicated Flag, Myc, and HA-Ub plasmids, immunoprecipitated with anti-Flag antibody, and probed with the indicated antibodies. Two independent experiments produced similar results. **A**–**H**. (*) indicates monoUb; (**) indicates diUb. Source data are provided as a Source Data file.

However, at this point, we cannot distinguish between mono-ubiquitylation on multiple sites and diubiquitylation on a single site.

We next determined the nature of the ubiquitylation on diUb-SAMPARP. If it resulted from a second monoUb linkage event, i.e. to a lysine on SAMPARP, then HA-Ub K0 would support formation, as it did for monoUb SAMPARP (shown in Fig. 4B). Alternatively, if it resulted from a diUb linkage event, then it would depend on a lysine in the HA-Ub protein. To address this, TNKS1/2 DKO cells were cotransfected with HA-Ub WT, K0, or various mutants containing a mutation at every lysine except the one indicated (K: 6,11,27,29,33,48,63), FlagDi19UIM or FlagRNF166 (FL), and MycSAMPARP. Lysates were immunoprecipitated with anti-HA antibody.

As shown in Fig. 4D, using WT HA-Ub, Di19UIM-induced monoUb (lane 2) and RNF166-induced diUb (lane 3). With K0 HA-Ub we observed a similar diUb band but, in addition, a strong faster migrating band that could be monoUb (lane 4). All the other HA-Ub mutants (except one, K11; lane 6) gave a pattern similar to K0. Only K11 resembled WT. To confirm the dependence of the WT pattern on a K11 linkage, we performed the reaction with HA-Ub containing a K11R mutation. As shown in Fig. 4E, K11 Ub resembles WT (lanes 7 and 8), whereas, K11R resembles K0 Ub (lanes 9 and 10) confirming that the K11 linkage is essential and sufficient for the wild-type pattern of ubiquitylation.

We were surprised to see the diUb band when using K0-Ub, since it should only support monoUb. We reasoned that the diUb form could be generated using endogenous WT Ub, which is very abundant[42]. To address this, we performed the analysis in an inducible shRNA ubiquitin knockdown cell line[43,44]. Cells were induced with dox for 48 hrs and then (24 hr prior to harvest) cotransfected with FlagDi19UIM or FlagRNF166 (FL), MycSAMPARP, and HA-Ub (K0 or K11; containing point mutations that render them resistant to the ubiquitin shRNA). The cell lysates were immunoprecipitated with anti-HA antibody. As shown in Fig. 4F, with K0 HA-Ub, Di19UIM induced monoUb (lane 8), as expected and as was shown above in Fig. 4B (lane 8). Moreover, with K0 HA-Ub, RNF166 induced only monoUb (Fig. 4F, lane 9). We do not detect any diUb form with K0-Ub (as we had in Fig. 4D and E when endogenous ubiquitin was present), indicating that the diUb form generated in Fig. 4D and E was likely due to endogenous ubiquitin and further that only WT or K11 Ub promote the diUb form.

Finally, as an alternative approach to validate the K11 linkage, we used a K11 linkage-specific deubiquitinating enzyme, Cezanne[45,46]. TNKS1/2 DKO cells were transfected with MycSAMPARP, FlagRNF166, and HA-Ub. Prior to gel analysis, the samples were incubated for 0 or 30 min with or without Cezanne at a concentration known to cleave only K11 linkage[45]. As shown in Fig. 4G (lane 4), the diUb form of SAMPARP was converted to the monoUb form following treatment with Cezanne.

The above analyses were done with the SAMPARP domain of tankyrase. We, thus performed a similar analysis using full-length TNKS1. TNKS1/2 DKO cells were transfected with FlagTNKS1, HA-Ub, and RNF166 and lysates immunoprecipitated with anti-Flag antibody. As shown in Fig. 4H, we detected an HA-Ub form of TNKS1 (likely monoUb based on its migration in the gel) (lane 8) with WT HA-Ub, that was converted to a slower migrating form by RNF166 (likely diUb) (lane 9). K11-Ub behaved like WT, promoting the diUb form (lane 10). However, K11R and K0 promote the monoUb form (lanes 11 and 12). They also show the diUb form, but this is likely due to endogenous ubiquitin as we showed above in Fig. 4E and F for SAMPARP. Together these data show that RNF166 or 114 induces K11-linked diubiquitylation of tankyrase.

## Determine the impact of RNF166 and K11-linked ubiquitylation on tankyrase

We have shown thus far that monoUbTNKS1 is a target for RNF166- or RNF114-mediated K11 diubiquitylation. K11 is an atypical ubiquitin linkage. While it has been shown to mark proteins for degradation, particularly in association with K48 in the form of K11/K48 mixed chains, it has also been shown to promote protein stabilization[47]. Indeed, a study on the innate immune pathway showed that K11, rather than cooperate with K48, could instead compete with K48-linked polyubiquitylation[48]. Here it was shown that competing E3 ligases (RNF5 and RNF26) controlled the stability of STING, a component of the cytosolic double-stranded DNA-sensing cGAS-cGAMP-STING pathway. Viral infection enhanced RNF5 interaction with STING, leading to STING K48-linked polyUb and proteasomal degradation to evade the innate immune response[49]. However, a competing E3 ligase, RNF26, promoted K11-linked ubiquitylation of STING to protect it from K48-linked polyUb and degradation[48].

We thus asked if RNF166 could compete with RNF146-mediated K48-linked polyubiquitylation and proteasomal degradation. TNKS1/2 DKO cells were transfected with FlagTNKS1 and HA-Ub, without additional E3 ligases or with RNF166 or RNF146 or both together, and immunoprecipitated with anti-Flag antibody. Although assessing the abundance of tankyrase is complicated by the different migration patterns, we observe effects of the E3 ligases on tankyrase levels. As shown in Fig. 5A, without an added E3 ligase, we observed mono-UbTNKS1 (lane 8). Introduction of RNF166 induced the diUb form of TNKS1 and led to TNKS1 stabilization (lane 7). By contrast, introduction of RNF146 led to polyUbTNKS1 and TNKS1 degradation (lane 9). When both RNF166 and RNF146 were introduced, RNF166 competed with RNF146 to promote diUbTNKS1 over polyUbTNKS1 and to prevent tankyrase degradation (lane 10).

To confirm the role of the specific ubiquitin linkages in this process, we performed the same analysis using (in place of WT HA-Ub) a mix of two specific linkages each with a unique tag: HA-K11-Ub and

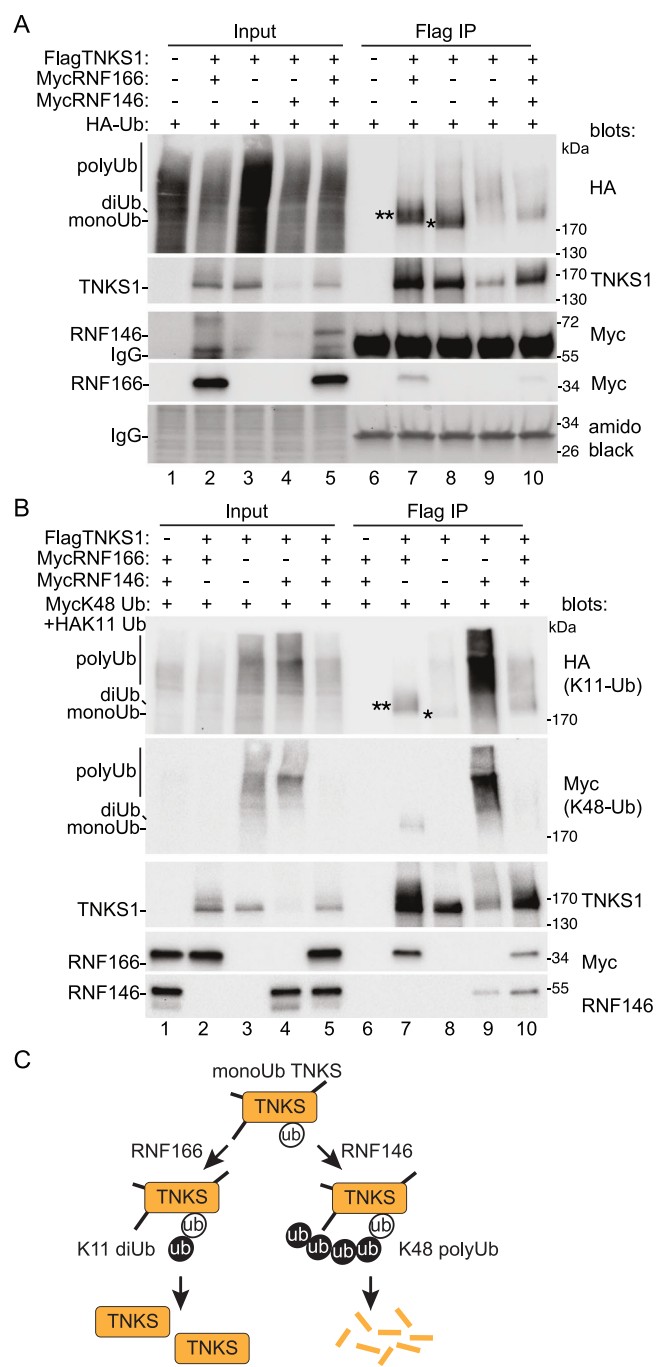

**Fig. 5 | RNF166 competes with RNF146. A** Immunoblot analysis of TNKS1/2 DKO HEK293T cells transfected with the indicated Flag, Myc, and HA-Ub plasmids, immunoprecipitated with anti-Flag antibody, and probed with the indicated antibodies or stained with amido black. At least three independent experiments produced similar results. **B** Immunoblot analysis of TNKS1/2 DKO HEK293T cells transfected with the indicated Flag, Myc, and HA-Ub plasmids, immunoprecipitated with anti-Flag antibody, and probed with the indicated antibodies. Two independent experiments produced similar results. **C** Model for competition between E3 ligases RNF166 and RNF146. **A, B** (*) indicates monoUb; (**) indicates diUb. Source data are provided as a Source Data file.

Myc-K48-Ub. TNKS1/2 DKO cells were transfected with FlagTNKS1 and the ubiquitin mix and analyzed by immunoprecipitation with anti-Flag antibody followed by blotting with anti-HA or -Myc antibodies. In the absence of added E3 ligases, we observed low levels of mono- and poly-K11UbTNKS1 (Fig. 5B, top panel, lane 8) and no K48 Ub (second panel),

likely due to degradation of that species. Introduction of RNF166 induced the K11diUbTNKS1 (as well as a low level of K48diUbTNKS), and led to TNKS1 stabilization (lane 7). Introduction of RNF146 led to K11 and K48 polyUb TNKS1 and the degradation of TNKS1 (lane 9), as described previously[24]. When both RNF166 and RNF146 were introduced, RNF166 competed with RNF146; it induced K11UbTNKS1, reduced K48-linked polyUbTNKS1, and promoted stabilization of TNKS1 (lane 10). Thus, as shown schematically in Fig. 5C, RNF166-mediated K11-linked diubiquitylation can compete with RNF146-mediated K48-linked polyubiquitylation to promote tankyrase stabilization over degradation.

## Determine the impact of RNF166 on TNKS1 binding partners

Tankyrase regulates the cellular levels of a number of vital proteins. Since these TBPs are subject to PAR-dependent RNF146-mediated degradation, we next asked if their stability could also be countered by RNF166. We considered that RNF166, TNKS1, and a TBP could form a ternary complex, where RNF166 binds the SAMPARP domain and the TBP binds the ankyrin domain, resulting in RNF166-mediated ubiquitylation and stabilization of the TBP (Fig. 6A, schematic). We took a proteomic approach to identify TBPs that could be in a ternary complex. We transfected TNKS1/2 DKO cells with FlagRNF166 alone versus FlagRNF166 plus TNKS1 (versus FlagBAP plus TNKS1 as a negative control) and performed Flag immunoprecipitation followed by mass spectrometry (Supplementary Data 1). We identified a number of TBPs in the Flag IP of RNF166 plus TNKS1, consistent with formation of ternary complexes. Unexpectedly, a subset of those TBPs bound to RNF166 even in the absence of TNKS1 and further, their interaction with RNF166 was increased upon expression of TNKS1 (Fig. 6A, Table). These TBPs are prime candidates for targets of RNF166-mediated ubiquitylation and stabilization.

We focused initially on the top hit, AMOT (Angiomotin), a regulator of YAP, an oncoprotein that is overexpressed in various cancers[31]. To confirm the association of AMOT, we transfected TNKS1/2 DKO cells with Flag RNF166 with or without TNKS1 and performed Flag IPs. As shown in Fig. 6B, endogenous AMOT was coimmunoprecipitated by RNF166 alone (lane 7) and it was enriched when TNKS1 was cotransfected (lane 8). We also observed a slower migrating form of AMOT, which could indicate ubiquitylation by RNF166. To measure ubiquitylation of AMOT we transfected FlagAMOT, TNKS1, HA-Ub, and MycRNF166 into TNKS1/2 DKO cells followed by IP with anti-Flag (Fig. 6C) or anti-HA (Fig. 6D) antibody. As shown in Fig. 6C, cotransfection of TNKS1 with AMOT led to a reduction in AMOT protein pull down (lane 9, top panel), likely due to TNKS1-mediated PARylation (lane 9, second panel) and degradation. However, when RNF166 was introduced along with TNKS1, AMOT protein level was rescued (lane 10, top panel), and we observed what appears to be diubiquitylation (although western blotting does not permit a precise determination of size) of AMOT (lane 10, third panel), as well as a concomitant reduction in AMOT PARylation (lane 10, second panel). These data suggest that RNF166-mediated ubiquitylation of AMOT protects it from PARylation by TNKS1 and subsequent degradation. We confirmed the RNF166/TNKS1-dependent diubiquitylation of AMOT using HA-Ubiquitin immunoprecipitation followed by immunoblotting to detect FlagAMOT (Fig. 6D, lane 10, top panel). Finally, we performed a similar analysis and show that we can detect diubiquitylation of endogenous AMOT (Fig. 6E, lane 8). Here we tested another target, tankyrase binding protein 1, TNKSBP1 (also known as TAB182)[14], which was highly enriched in the TNKS1/RNF166 immunoprecipitate (Supplementary Data 1). As shown in Fig. 6E, lane 8, endogenous TAB182 shows increased ubiquitylation upon cotransfection of TNKS1 and RNF166.

To determine if RNF166 ubiquitylates AMOT through a K11 linkage, TNKS1/2 DKO cells were cotransfected with FlagAMOT, MycRNF166, TNKS1, and HA-Ub WT, K11, K11R, or K0 and immunoprecipitated with Flag antibody. As shown in Fig. 6F, monoUbAMOT

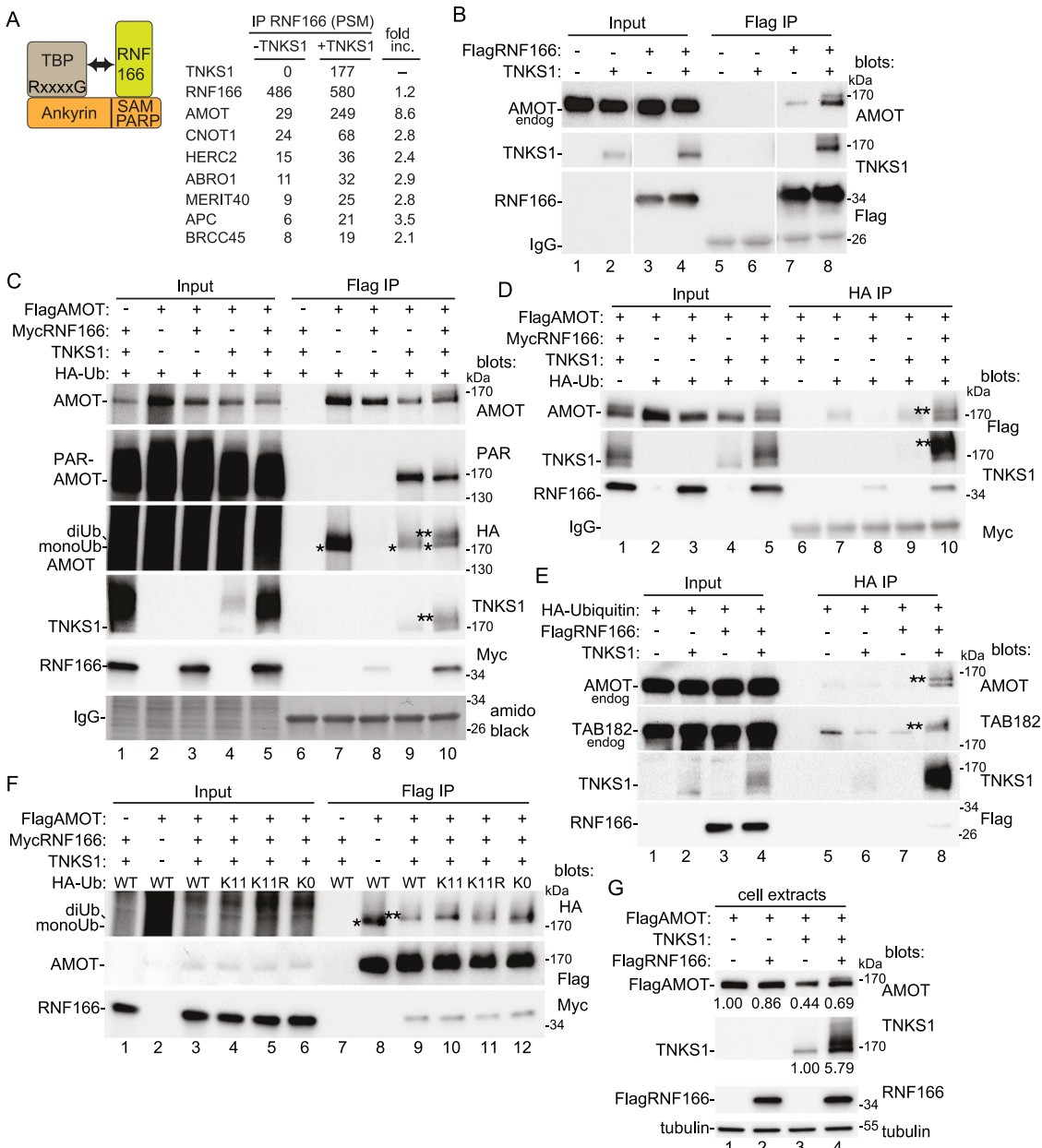

**Fig. 6 | Analysis of the effect of RNF166 on tankyrase binding proteins.**
**A** Schematic of a ternary complex. Table of the proteomics analysis of proteins that immunoprecipitate with RNF166 alone versus RNF166 + TNKS1. Fold increase was calculated using the PSM ratio between the two conditions. **B** Immunoblot analysis of TNKS1/2 DKO HEK293T cells transfected with FlagRNF166 and TNKS1, immunoprecipitated with anti-Flag antibody, and probed with the indicated antibodies. At least two independent experiments produced similar results. **C** Immunoblot analysis of TNKS1/2 DKO HEK293T cells transfected with FlagAMOT, MycRNF166, HA-ubiquitin, and TNKS1, immunoprecipitated with anti-Flag antibody, and probed with the indicated antibodies or stained with amido black. At least two independent experiments produced similar results. **D** Immunoblot analysis of TNKS1/2 DKO HEK293T cells transfected with FlagAMOT, MycRNF166, HA-ubiquitin, and TNKS1, immunoprecipitated with anti-HA antibody, and probed with the indicated antibodies. At least two independent experiments produced similar results.

**E** Immunoblot analysis of TNKS1/2 DKO HEK293T cells transfected with FlagRNF166, TNKS1, and HA-Ub plasmids, immunoprecipitated with anti-HA antibody, and probed with the indicated antibodies. Two independent experiments produced similar results. **F** Immunoblot analysis of TNKS1/2 DKO HEK293T cells transfected with FlagAMOT, MycRNF166, HA-ubiquitin, and TNKS1plasmids, immunoprecipitated with anti-Flag antibody, and probed with the indicated antibodies. At least two independent experiments produced similar results.
**G** Immunoblot analysis of TNKS1/2 DKO HEK293T cells transfected with FlagAMOT, FlagRNF166, and TNKS1, and probed with the indicated antibodies. Protein levels relative to tubulin and normalized to the control are indicated below the blots and are representative of two independent experiments. At least two independent experiments produced similar results. **C**–**F** (*) indicates monoUb; (**) indicates diUb. Source data are provided as a Source Data file.

was detected when AMOT alone was transfected (lane 8). Addition of TNKS1 and RNF166 efficiently converted monoUbAMOT to higher molecular weight forms. Although western blotting does not permit a precise determination of size, their migration is consistent with diUb for WT (lane 9) or K11 Ub (lane 10). Introduction of K11 (lane 11) or K0 (lane12), led to the appearance of the monoUb form, but

some diUbAMOT remained likely due to endogenous ubiquitin as was shown above in Fig. 4 for TNKS1. To address this, we repeated the analysis in the inducible shRNA ubiquitin knockdown cell line. As shown in Supplementary Fig. 3A, we can now more clearly distinguish the migration pattern between K11 HA-Ub (lane 10) and K11R HA-Ub (lane 11).

The data thus far suggest that RNF166-induced K11 ubiquitylation could prevent the TNKS1-mediated degradation of AMOT. To address this directly we monitored the levels of AMOT in response to TNKS1 and RNF166 in TNKS1/2 DKO cells. As shown in Fig. 6G, transfection of RNF166 alone had a minimal effect on AMOT levels (lane 2). TNKS1 alone led to loss of AMOT (lane 3) as expected because TNKS1 PARylates AMOT and promotes RNF146-mediated degradation. When RNF166 was introduced along with TNKS1, it partially rescued AMOT levels (lane 4). The rescue was particularly striking since TNKS1 was increased over 5-fold (lane 4) yet it did not promote degradation of AMOT. Thus, RNF166 protects AMOT from TNKS1-mediated degradation.

### Identify E3 ligases that target PARylated TNKS1 for ubiquitylation

We have shown that RNF166 promotes K11-linked ubiquitylation of TNKS1, which can compete with K48-linked ubiquitylation and promote TNKS1 stabilization. The N-terminal RING-C2HC domain catalyzes the reaction, while the C-terminal Di19-UIM stabilizes the target, monoUbTNKS1. We have shown that stabilization and ubiquitylation of monoUbTNKS1 is sensitive to TNKSi and does not occur on catalytically dead TNKS1. However, it is not clear why the catalytic activity of TNKS is important. Unlike RNF146, which contains a WWE-PAR-binding domain and binds PARylated proteins, RNF166 lacks a PAR-binding domain and does not bind to PARylated protein, (shown for PARP1; see Fig. 1I). We consider the possibility that the generation of mono-UbTNKS1 (the target for RNF166-mediated K11 ubiquitylation) requires a catalytically active TNKS1. Thus, we hypothesize that another E3 ligase (perhaps a PAR-binding one) is required to generate the target (Fig. 7A, schematic).

The most likely candidate is RNF146. To determine if RNF146 was essential to generate monoUbTNKS1 we performed the analysis in RNF146 KO cells. We cotransfected Flag-Di19UIM, HA-Ub, and TNKS1 into RNF146 KO cells and lysates were immunoprecipitated with anti-HA antibody. As shown in Fig. 7A, Di19UIM induced monoHA-Ub TNKS1 in the absence of RNF146. Thus, there may be other PAR-binding E3 ligases that target TNKS1.

In theory, any protein with a PAR binding motif and a ubiquitin E3 ligase domain could be a candidate for a PAR-dependent E3 ligase that targets tankyrase[27,50]. At least seven potential PAR-binding E3 ligases can be identified (Fig. 7B): four have a WWE domain with a RING (RNF146, DTX1, DTX2, and DTX4), two have WWE with a HECT (TRIP12 and HUWE1), and one has PBZ with a RING (CHFR)[51]. Three (RNF146, TRIP12, and CHFR) have been shown to target PARP1 for ubiquitylation and degradation[26,28,50,52]. Only one (RNF146) has been shown to act on tankyrase[24,25].

First, we asked if the E3 ligases bind TNKS1. TNKS1 was transfected into TNKS1/2 DKO cells along with Flag-tagged BAP, RNF146, CHFR, DTX1, DTX2, or DTX4 and immunoprecipitated with anti-Flag antibody. As shown in Fig. 7C, the E3 ligases coimmunoprecipitated TNKS1 (lanes 8-12). Indeed, despite low levels of expression for the DTX plasmids, all showed a robust immunoprecipitation of TNKS1 (lanes 10−12), similar to RNF146 and CHFR. We focused first on CHFR. FlagCHFR or RNF166 was cotransfected with TNKS1 into TNKS1/2 DKO cells. In contrast to RNF166, which leads to stabilization of TNKS (Fig. 7D, lane 2), CHFR led to the loss of TNKS1 (lane 3) that was rescued with TNKSi (lane 4). Coimmunoprecipitation of CHFR with TNKS1 was also reduced with TNKSi (lanes 7 and 8). These data suggest that CHFR may be acting more like RNF146 to stimulate degradation of TNKS1. To address this, we transfected TNKS1/2 DKO cells with FlagTNKS1, HA-Ub, and MycRNF146 or CHFR and performed Flag immunoprecipitation. As shown in Fig. 7E, in the absence of E3 ligases we detected monoUbTNKS1 (lane 8). Addition of RNF146 or CHFR-induced poly-UbTNKS1 (lanes 9 and 11), that was stabilized by treatment with the proteasome inhibitor MG132 (lanes 10 and 12). Thus CHFR, like

RNF146, targets TNKS1 for polyubiquitylation and degradation by the proteasome.

We next turned to the DTX proteins to determine if their interaction with TNKS1 depended on TNKS1 catalytic activity. As shown in Fig. 7F, DTX1, DTX2, and DTX4 coimmunoprecipitated TNKS1 (lanes 3, 5, and 7) and coimmunoprecipitation was abrogated by treatment with TNKSi (lanes 4, 6, and 8). To measure a role for the DTX proteins in ubiquitylation of TNKS1, we cotransfected MycDTX1, 2, or 4, or MycRNF166 with FlagTNKS1 and HA-Ub into TNKS1/2 DKO cells, and performed Flag immunoprecipitation. As shown in Fig. 7G, in the absence of E3 ligases we detected monoUbTNKS1 (lane 8). RNF166 induced the diUb form (lane 9). By contrast, DTX1, 2, and 4 did not induce additional ubiquitylation, but rather they induced stabilization of the monoUbTNKS1 (lanes 10, 11, and 12). Finally, we tested the remaining two candidate E3 ligases, HUWE1 and TRIP12, which each contain a WWE PAR-binding domain along with a HECT E3 ligase domain. As shown in Fig. 7H, a slower migrating form of TNKS1 (consistent with ubiquitylation) coimmunoprecipitated with HUWE1 (lane 9), dependent on TNKS1 catalytic activity (lane 10). We did not detect an interaction with TRIP12. Thus, we detected an interaction between tankyrase and all the E3 ligases (Fig. 7B) except TRIP12, dependent on its catalytic activity. To determine if HUWE1 stimulates ubiquitylation of TNKS1 we contransfected FlagHUWE1, TNKS1, and HA-Ub into TNKS1/2 DKO cells, and performed HA immunoprecipitation. As shown in Fig. 7I, HUWE1 stimulated ubiquitylation of TNKS1.

## Discussion

Tankyrases have a remarkably broad range of interacting partners and cellular functions. They undergo auto PARylation and PARylate many of their binding partners. The main avenue for regulation of tankyrase protein level is through the PAR-dependent E3 ligase RNF146, which promotes K48-linked polyubiquitylation and degradation of PARylated tankyrase and its PARylated partners. Here we describe an antagonist to that degradation. We show that the RING-UIM E3 ligases RNF166 and RNF114 bind monoUbTNKS1 to prevent RNF146-mediated K48-linked polyubiquitylation and degradation and to promote K11-linked diubiquitylation and TNKS1 stabilization (see model Fig. 5C).

In order to effectively compete with RNF146, RNF166 must be able to gain access to the RNF146 target: PARylated tankyrase. However, RNF166 does not contain a known PAR-binding domain and does not bind to other PARylated proteins, such as PARP1. Nonetheless, RNF166-mediated stabilization and ubiquitylation of tankyrase depends on TNKS1 catalytic activity. To elucidate the mechanism, we characterized the subdomains of RNF166. We showed that the Di19 domain was necessary and sufficient for binding tankyrase and was unaffected by TNKSi. Di19 binding alone did not induce tankyrase level/modification. However, addition of just 17 amino acids (in the form of the UIM domain) to Di19 led to a TNKSi-sensitive increase in TNKS1 protein levels and the appearance of a monoUb species. We hypothesize that TNKS is monoubiquitylated by a PAR-binding E3 ligase, hence the dependence on catalytic activity.

We propose that the Di19 and the UIM domains act together as a reader to bind and stabilize monoUbTNKS. Di19 alone detects some unique feature of TNKS1. A recent study identified RNF114 as a mono-ADPr (MAR) reader and showed that it binds directly to MAR using its Di19 domain[41]. We showed here that the TNKS SAMPARP domain is MARylated. Thus, the Di19 could bind to MAR on SAMPARP. Such a mark might be conferred by degradation of PAR by the PAR glycohydrolase (PARG) or through the action of a MAR-transferase[53]. Once the Di19 binds, the UIM could then bind to a nearby ubiquitin and block elongation of ubiquitin chains[54–56]. The RING-C2HC domain could then promote K11-diubiquitylation of TNKS1 and (its partners) to promote alternative pathways (Fig. 7J, Model).

What is the origin of the monoUb TNKS1 species that the Di19-UIM binds to and stabilizes? In our experiments this species does not

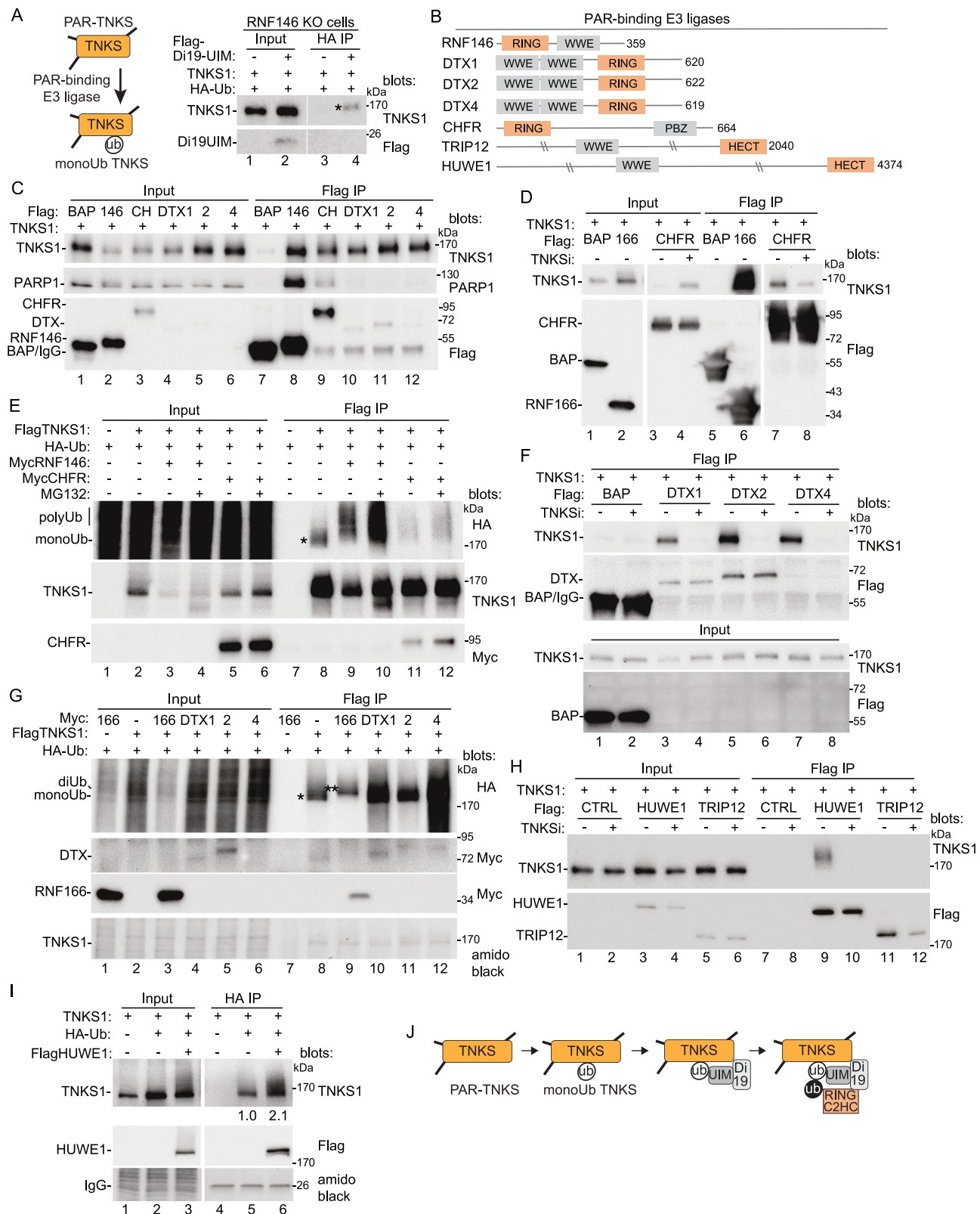

require addition of an exogenous E3 ligase. Thus, there may be a low constitutive level of this species in cells that can be stabilized by RNF166. Mono- and polyubiquitylation of proteins are considered separate steps. The process can use distinct E3s, the same E3s (with distinct E2s), or E2s[57-59]. Since appearance of this species depends on TNKS1 catalytic activity, the PAR-binding E3 ligase RNF146 is a good candidate to generate monoUbTNKS1. However, we found that the

Di19UIM could induce monoUb TNKS even in RNF146 KO cells, suggesting that while RNF146 may do so, other E3 ligases may be capable. To date, RNF146 is the only PAR-binding E3 ligase shown to act on tankyrase. We tested six other candidate E3s that contained PAR-binding domains. All except one (TRIP12), bound tankyrase and promoted ubiquitylation dependent on TNKS1 catalytic activity. CHFR and HUWE1 induced polyUb, similar to RNF146. The other three (DTX1, 2,

**Fig. 7 | Identify E3 ligases that target PARylated TNKS for ubiquitylation.**
**A** Schematic diagram indicating a PAR-binding E3 ligase could promote mono-ubiquitylation of TNKS. Immunoblot analysis of RNF146 KO HEK293A cells transfected with FlagDi19UIM, TNKS1, and HA-Ub, immunoprecipitated with anti-HA antibody, and probed with the indicated antibodies. At least two independent experiments produced similar results. **B** Schematic presentation of PAR-binding E3 ligases with their binding and E3 catalytic domains. **C** Immunoblot analysis of TNKS1/2 DKO HEK293T cells transfected with the indicated Flag plasmids and TNKS1, immunoprecipitated with anti-Flag antibody, and probed with the indicated antibodies. At least three independent experiments produced similar results. **D** Immunoblot analysis of TNKS1/2 DKO HEK293T cells transfected with the indicated Flag plasmids and TNKS1, with and without TNKSi, immunoprecipitated with anti-Flag antibody, and probed with the indicated antibodies. At least two independent experiments produced similar results. **E** Immunoblot analysis of TNKS1/2 DKO HEK293T cells transfected with FlagTNKS1, MycRNF146 or MycCHFR, and HA-Ub, with and without MG132, immunoprecipitated with anti-Flag antibody, and probed with the indicated antibodies. At least three independent experiments produced similar results. **F** Immunoblot analysis of TNKS1/2 DKO HEK293T cells

transfected with the indicated Flag plasmids and TNKS1, treated with and without TNKSi, immunoprecipitated with anti-Flag antibody, and probed with the indicated antibodies. At least two independent experiments produced similar results. **G** Immunoblot analysis of TNKS1/2 DKO HEK293T cells transfected with the indicated Myc plasmids, TNKS1, and HA-Ubiquitin, immunoprecipitated with anti-Flag antibody, and probed with the indicated antibodies or stained with amido black. At least two independent experiments produced similar results. **H** Immunoblot analysis of TNKS1/2 DKO HEK293T cells transfected with the indicated Flag plasmids and TNKS1, immunoprecipitated with anti-Flag antibody, and probed with the indicated antibodies. At least two independent experiments produced similar results. **I** Immunoblot analysis of TNKS1/2 DKO HEK293T cells transfected with Flag HUWE1, TNKS1 and HA-Ub plasmids, immunoprecipitated with anti-HA antibody, and probed with the indicated antibodies or stained with amido black. Protein levels relative to tubulin and normalized to the control are indicated below the blots. Two independent experiments produced similar results. **J** Model for TNKS ubiquitylation by RNF166. **A, E, G** (*) indicates monoUb; (**) indicates diUb. Source data are provided as a Source Data file.

and 4) did not induce diUb or polyUb, but rather led to stabilization of monoUb TNKS1 and thus, could be candidates for E3 ligases that monoubiquitylate PARylated TNKS. Interestingly, a recent study showed that DTX E3 ligases can ubiquitylate ADP-ribosylated proteins by attaching ubiquitin to the protein-linked ADPr modification[60]. Future experiments will determine if this is the case for DTX-mediated ubiquitylation of TNKS1.

We showed that RNF166 can compete with RNF146 to prevent degradation of TNKS1. There are a number of examples of competing E3 ligases in innate immunity, such as the one described above for RNF5 and RNF26 in the DNA-sensing STING pathway[48]. The RING-E3 ligases can positively and negatively regulate innate signaling pathways through ubiquitylation[33]. In addition to the DNA-sensing STING pathway, there is the RNA-sensing RIG-I/MDA5-MAVS/TRAF3 pathway, and the RNA/DNA sensing Toll Like Receptor (TLR)-mediated TRIF/TRAF3 pathway[33]. RNF166 impacts the TRAF3/TRAF2 pathway[61], and RNF114 and 125 the RIG-I/MDA5-MAVS pathway[62,63]. Tankyrase has been implicated in these same innate immune response pathways: tankyrase attenuates TLR signaling through RNF146-mediated degradation of its target 3BP2[64] and inhibits the innate immune antiviral response by PARylating MAVS and promoting its RNF146-mediated ubiquitylation and degradation[65]. Future experiments will determine if there is cross-talk/competition between RNF146, tankyrase, and the RING-UIM E3s in innate signaling pathways.

We show that RNF166 binds to the SAMPARP domain of TNKS1. This is the first example of a TBP (other than tankyrase itself) that binds SAMPARP and not the ankyrin domain. Such binding offers an opportunity for the formation of a ternary complex, as shown for AMOT, TNKS1, and RNF166. We showed that in this setting PARylation of AMOT was reduced. RNF166 binding to SAMPARP may limit the ability of TNKS1 to PARylate a bound TBP, which would limit its interaction with RNF146. At the same time RNF166 (by binding and capping the monoUb and by K11 diubiquitylation) could block elongation of ubiquitin chains. Overall, this could provide an effective counter to RNF146-mediated degradation of AMOT, resulting in AMOT stabilization. AMOT suppresses the oncogenic function of the transcriptional activator YAP. Tankyrase inhibitors stabilize AMOT and suppress YAP oncogenic functions[31,66]. Thus, RNF166-mediated stabilization of TNKS1/AMOT, could potentially suppress the oncogenic function of YAP. We identified a number of TBPs that may be in a ternary complex with RNF166 and TNKS1 and demonstrated increased ubiquitylation for one, TAB182. Future experiments will determine if their fate is similar to AMOT.

The ability of RNF166 (and RNF114) to stabilize TNKS1 by blocking degradation, in itself, offers a window of regulation. Additionally, the K11-linked ubiquitylation may promote protein-protein interactions

and reroute TNKS1 to other pathways including autophagy, innate immunity, and cell cycle regulation[47]. Our study provides insights into mechanisms of tankyrase regulation and may offer strategies for targeting tankyrases in human disease.

## Methods

### Cell lines

The following cell lines were supplemented with 10% DBS and grown in standard conditions: HEK293T (ATCC CRL-1573), U2OS (ATCC HTB-96), HEK293T TNKS1/2 DKO[16], HEK293A WT and RNF146 KO[67] (provided by Dr. Junjie Chen), Super HeLa[68] (provided by Dr. Joachim Lingner), hTERT RPE-1 PARP1$^{-/-}$/PARP2$^{-/-}$ cells[38] (provided by Dr. Keith Caldecott), and U2OS shUb[43] (provided by Dr. Niels Mailand).

### Cell transfection and treatment

The cells were seeded in 6 well plate and treated the next day using 2 μg of plasmid and 4 μL of lipofectamine 3000 (Invitrogen) for 24 h according to the manufacturer's instructions. The cells were harvested in cold PBS for the 293 T lines or with trypsin for all other cell lines. Tankyrase inhibitor #8 (TNKSi) (Chembridge Corporation, MolPort-000-222-699) was used at a concentration (10 μM) that does not inhibit PARP1 or 2[36]. Olaparib (Sigma) was used at 1 μM for 24 h. For U2OS shUB transfections, cells were treated with doxycycline (0.5 μg/ml) for 48 h and transfected 24 h prior to harvesting. MG132 (Fisher) was added at 10 μM for 4 h prior cells harvesting. For harvesting lysing samples for quantification with anti-PAR antibody, PJ34 (VWR) was added at 8 μM.

### Plasmids

Plasmids used in this study are listed Supplementary Data 2. The following plasmids were used: MycTRF1[69]; pLPCFlagTNKS1[18]; TNKS.WT (TT20.WT)[5]; TNKS.CD (TT20.PD; PARP dead)[70]; and 3XFlag-BAP (Sigma).The following plasmids were provided: pCMV-3XF-RNF166[39](provided by Dr. Ramnik Xavier); pcDNA3.1-Flag-TRIP12-WT[50] (provided by Dr. Matthias Altmeyer); and pcDNA3-Myc- DTX1, DTX2 and DTX4[71] (provided by Dr. Danny Huang). The following plasmids were obtained from Addgene: RNF114 (58295, from Francesca Capon); RNF125 (122045, from Ze'ev Ronai); RNF138 (78920, from Michael Hendzel); RNF146 (132610, from Wenqing Xu); CHFR (61853, from Jonathon Pines); AMOT (32828, from Kunliang Guan); and HUWE1 (187155, from Eric Fischer). Unless already fused to a 3XFlag or a 3Xmyc tag, the cDNA from the plasmids cited above were subcloned to express N-ter 3XMyc or 3XFlag tagged proteins, except for HUWE1, which was fused to a C-ter 3XMyc tag (pDARMO) vector. Plasmid DNA sequences were modified using PCR amplification, enzymatic restriction, and ligation, or site-directed mutagenesis

(Agilent) and (NEB) or DNA assembly (NEBuilder® HiFi DNA Assembly Master Mix). RNF166 and TNKS1 truncations and mutants were generated using primers listed Supplementary Data 2. The following ubiquitin plasmids were obtained from Addgene: pRK5-HA-Ubiquitin WT (17608, from Ted Dawson); K0 (17603, from Ted Dawson); K6 (22900, from Sandra Weller); K11 (22901, from Sandra Weller); K27 (22902, from Sandra Weller); K29 (22903, from Sandra Weller); K33 (17607, from Ted Dawson); K48 (17605, from Ted Dawson); K63 (17606, from Ted Dawson); and K11R (121154, from Josef Kittler). pRK5-Myc-Ubiquitin-K48 was cloned using primers listed in Supplementary Data 2.

## Protein extraction and immunoprecipitation

Proteins were extracted by resuspending the cell pellets for 1 h on ice in TNE buffer [10 mM Tris (pH 7.8), 1% Nonidet P-40, 0.15 M NaCl, 1 mM EDTA, 2.5% protease inhibitor cocktail (Sigma, P8340), 1 μM of PARGi (Sigma)]. NEM (Sigma) was added for ubiquitin related experiments at 20 mM. The lysates were pelleted at 10,000 g for 10 min and the supernatants were used to determine the protein concentration using Bradford assay (Bio-Rad). After a preclearing step with Protein G Sepharose (Sigma), equal amounts of proteins were then incubated with anti-Flag Beads (Sigma), anti-Myc Beads (Sigma) or anti-HA Beads (Sigma) during 2 h at 4 °C under agitation. For IP of endogenous proteins, protein extract (4.6 mg) was incubated for 2 h with α-TNKS1 465[5], α-RNF114 (Sigma) or rabbit IgG control (Cell Signaling Technology) and then Protein G beads were added for 1 h. The beads were washed at least three times with 1 mL of TNE buffer. The samples were denatured in Laemmli buffer for 5 min at 95 °C.

For detection of mono-ADP-ribose immunoprecipitations were performed under denaturing conditions. Pelleted cells were lysed using SDS lysis buffer (2% SDS, 0.9% Nonidet P-40, 9 mM Tris pH 8.0, 135 mM NaCl, 0.9 mM EDTA) and heated at 70 °C for 10 min. After a ten-fold dilution with TNE buffer containing PIC, PARGi and NEM, the samples were sonicated 1 min using a Diagenode bioruptor device (10 sec on / 10 sec off) and then centrifuged 15 000 g for 15 min at 4 °C. The supernatants were incubated with anti-Flag beads for 2 h. The beads were then washed three times with TNE and one time with 1 M NaCl lysis buffer (20 mM Tris pH 8.0, 1 M NaCl, 1% Triton X-100, 2 mM EDTA, 0.2% SDS). The samples were resuspended in Laemmli buffer and heated at 70 °C for 10 min.

## Cezanne deubiquitinase assay

Purified Cezanne catalytic domain (UbpBio, H4200) was used according to the UbiCREST assay[45]. Briefly, after the last anti-HA beads washing, beads were incubated with or without Cezanne (previously incubated with the DUB dilution buffer: 25 mM Tris pH 7.5, 150 mM NaCl, 10 mM DTT) at a final concentration of 0.2 μM in presence of 1X DUB reaction buffer (50 mM Tris pH 7.5, 50 mM NaCl, 5 mM DTT) in a final volume of 30 μL. After 30 minutes at 37 °C under shaking at 1000 rpm, the reaction was stopped by addition of Laemmli buffer and denatured at 70 °C for 10 min.

## Immunoblot analysis

Protein samples were loaded using precast gels (Bio-Rad), subjected to SDS-PAGE, and transferred onto nitrocellulose membrane using wet transfer of 100 V for 1 h. The membranes were stained with amido black and incubated with primary antibodies (listed Supplementary Data 2), including anti-TNKS 762 and 763[72] and anti-TAB182[14], followed by HRP-coupled secondary antibodies. For detecting mono-ADP-ribose, anti-MAR antibody (AbD43647) BioRad was incubated with BiSpyCatcher2-HRP (TZC002P) BioRad at a ratio 10:0.8 (v/v) for 1 hour before a 500-fold dilution in 5% non-fat milk PBST. The signal was acquired using ECL (Fisher) and the ChemiDoc MP imaging system (Bio-Rad).

## Protein pull-down for mass spectrometry

HEK293T cells were seeded in 10 cm dish and transfected the next day using 8 μg of plasmids, and the cells were then subjected to coimmunoprecipitation as described above using the Flag-Beads. After washing the beads with TNE buffer, the proteins bound to the Flag antibodies were eluted using 400 μg/mL of 3xFlag peptide (Sigma, F4799) during 30 min at 4 °C under agitation. Eluates were denatured in Laemmli buffer during 5 min at 95 °C and submitted for mass spectrometry.

## Mass Spectrometry sample preparation

Samples were reduced by DTT during 1 h at 57 °C, alkylated with IAA at RT for 45 min and loaded in a NuPAGE® 4-12% Bis-Tris Gel 1.0 mm (Life Technologies). After a migration of 20 min at 200 V, bands were stained using GelCode Blue Stain Reagent (Thermo) and then destained, excised and dehydrated with 50% methanol and 100 mM ammonium bicarbonate solution. The proteins were incubated with acetonitrile and 5% formic acid, concentrated using SpeedVac, and digested overnight at RT with 500 ng of modified trypsin (Promega) in 100 mM ammonium bicarbonate. Peptides were then loaded onto equilibrated microspin Harvard apparatus (Millipore) using a microcentrifuge, rinsed three times with 0.1% TFA and eluted with 40% acetonitrile in 0.5% acetic acid followed by the addition of 80% acetonitrile in 0.5% acetic acid. The organic solvent was removed using a SpeedVac concentrator and samples were reconstituted in 0.5% acetic acid.

## Mass spectrometry analysis

Ten percent of each sample was individually subjected to a liquid chromatography separation using the autosampler of an EASY-nLC 1200 HPLC (Thermo Fisher Scientific). Peptides were gradient eluted during 1 h using solvent A (2% acetonitrile, 0.5% acetic acid) and solvent B (80% acetonitrile, 0.5% acetic acid) into an Orbitrap Eclipse mass spectrometer (Thermo Scientific). High-resolution full MS spectra were acquired with a resolution of 120,000, an AGC target of 4e5, with a maximum ion time of 50 ms, and scan range of 400 to 1500 m/z. All MS/MS spectra were collected using the following instrument parameters: resolution of 30,000, AGC target of 2e5, maximum ion time of 200 ms, one microscan, 2 m/z isolation window, fixed first mass of 150 m/z, and normalized collision energy (NCE) of 27.

## Data processing for mass spectrometry

MS/MS spectra were searched against a Uniprot human database using Sequest within Proteome Discoverer 2.5. The fold change between two conditions was calculated using the ratio of PSM + 5 for each condition. Potentially relevant proteins were selected for having both fold changes ≥1.85 (shown in green in the Supplementary Data 1) for RNF166 + CTRL vs. BAP + CTRL conditions and for RNF166 + TNKS1 vs. RNF166 + CTRL conditions.

## Statistical analysis

Statistical analysis was performed using Prism 10 software. Data are shown as mean ± SEM. Student unpaired $t$ test was applied. $P < 0.05$ values were considered significant: *, $P \leq 0.05$; **, $P \leq 0.01$.

## Reporting summary

Further information on research design is available in the Nature Portfolio Reporting Summary linked to this article.

# Data availability

All data supporting the findings of this study are available within the article and its Supplementary Information files. Data are available from the corresponding author upon request. The proteomics data discussed in this study have been deposited to ProteomeXchange (hVps://www.proteomexchange.org/) via MassIVe partner repository

and are accessible through the accession number PXD042595. Source data are provided with this paper.

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

## Acknowledgements

We thank Katherine Ring and Samantha Sze for helpful discussions and critical reading of the manuscript. We thank Paige Sherman for technical assistance. We thank Keith Caldecott, Junjie Chen, and Neils Mailand for providing cell lines and Ramnik Xavier, Matthias Altmeyer, and Danny Huang for plasmids. Research reported in this publication was supported by the National Institute of General Medicine Sciences of the National Institutes of Health under award numbers R01GM129780 and R01GM141292 to S.S. We thank Beatrix Ueberheide and the Proteomics Laboratory at New York University School of Medicine, supported partly by the Laura and Isaac Perlmutter Cancer Center Support grant P30CA016087 from the National Cancer Institute. The Orbitrap mass spectrometer was purchased with a shared instrumentation NIH grant 1S10OD010582-01A1.

## Author contributions

J.P. and S.S. conceived the experimental design, analyzed the data, and wrote the manuscript. J.P. performed all experiments.

## Competing interests

The authors declare no competing interests.
