## [Peer Review File · Nature Communications]

Multiple E3 ligases control tankyrase stability and functionREVIEWER COMMENTS

Reviewer #1 (Remarks to the Author):

This is an exciting study that offers important new insights into the regulation of the stability of tankyrase and its substrate/client proteins in a wide range of biological processes. Through a comprehensive set of experiments building on initial observations by Li et al., 2017, the authors show that two E3 ligases of the RING-UIM family, RNF166 and RNF114, promote the K11-linked di-ubiquitylation of tankyrase and AMOT, a model tankyrase binder. Di-ubiquitylation competes with RNF146-mediated poly-ubiquitylation, thereby giving rise to the stabilisation of tankyrase and a subset of its substrates. The authors also propose a number of E3 ubiquitin ligases responsible for PARylation activity-dependent mono-ubiquitylation of tankyrase and its clients; these mono-ubiquitylated products could serve as substrates for the subsequent modification to generate K11-linked di-ubiquitin. The findings substantially further our understanding of the molecular events downstream of tankyrase-dependent PARylation. Given the diverse links of tankyrase to various (in part disease-relevant) biological processes, the findings are of broad relevance.

The study is comprehensive and well-conducted. The experiments are thorough. Please see below for a number of relatively small suggestions that could further improve an already strong manuscript.

Specific points

p. 3: Referring to TNKS/TNKS2 collectively as tankyrase is helpful, but I suggest to not use the gene symbol "TNKS" to do so (end of penultimate paragraph), as this is reserved for tankyrase 1. Especially as paralogue-specific functions are emerging, clearly distinguishing paralogues where necessary remains important.

p. 3, last paragraph: "Colorectal cancer screen" is vague. The Wnt/beta-catenin pathway is described not entirely correctly. AXIN degradation occurs late and is not responsible for the acute activation of the pathway. Instead, the consensus in the field is that the destruction complex is inactivated upon Wnt stimulation, which initiates its remodelling to a Wnt signalosome. In mammalian cells, AXIN1 may not be the concentration-limiting factor of the destruction complex, but APC (PMID: 28810742).

p. 5, paragraph 2, and subsequent: "TBS" is rather unconventional: can "TBM" be used? Can you also point out that these are "candidate" or "potential" TBMs?

Figure 1A: Can all eight residues of the potential tankyrase binding motifs be spelled out? This way, it will be evident that the motifs themselves are likely weak.

p. 5, paragraph 2: What is the rationale of using the tankyrase inhibitor at such a high concentration?

p. 6, paragraph 2: Referring to Figure 1E, the authors suggest that the stabilising effect of RNF166 and RNF144 on tankyrase is affected by tankyrase inhibition. This does not appear to be the case, at least in Figure 1E.

p. 6, paragraph 3: "stimulation of tankyrase [levels]" (to distinguish this from activity)

Figure 1G: It is remarkable to obtain such a clear Western blot signal for PAR from a cell lysate. Out of curiosity, do the authors know the identity of the strong single band that appears to accumulate in the anti-PAR blot upon tankyrase inhibition?

Figure 1H: The authors could quantify additional signals, for example tankyrase levels upon tankyrase inhibition or the extent of PARylation, if feasible.

Figure 2D: For the sake of completeness, the schematic of the deltaUIM mutant variant could be added.

p.9, paragraph 1: "This is the first example of a tankyrase binding protein [other than tankyrase itself] that binds to the SAMPARP ...". Recent studies show extensive self-interactions of SAMPARP. The same applies to p. 18, paragraph 2.

Figure 3B: What is the explanation for the second band appearing for TRF1, around 40 kDa? It is not observed in panel A.

p.9, last paragraph: "Note that we also detect some unmodified TNKS, which is likely due to its ability to oligomerize ..." - The clarity of this statement could be improved by explaining that unmodified tankyrase could potentially heteropolymerise with modified tankyrase in the experiment.

Figure 4B: Could the anti-Myc panel be expanded to include higher-molecular-weight species, similar to the equivalent panel in Figure 4C? This will facilitate comparison of the experiments.

Figure 4C: Co-expression of MycSAMPARP with HA-Ub and either RNF114 or RNF166 gives rise to slower-migrating species likely corresponding to di-ubiquitylated forms of MycSAMPARP. An additional control condition without the expression of any E3 ligase (just MycSAMPARP and HA-Ub) would have been desirable.

p. 10, paragraph 3: As the authors do not distinguish between mono-ubiquitylation on multiple sites and di-ubiquitylation on a single site, it would be helpful to interpret Figure 4C accordingly at this stage in the manuscript. This would also nicely lead into the subsequent paragraph.

Figure 4D: Can the anti-Myc input panel be extended towards higher molecular weight to match the corresponding IP panel?

p. 12, paragraph 3 and Figure 5A: The authors state that expression of RNF166 leads to tankyrase stabilisation. However, this cannot be appreciated from the figure, in either the IP or the input samples. As levels are compared, the input samples require a loading control for analysis.

p. 13, paragraph 1 and Figure 5B: Assessing the abundance of tankyrase is complicated by the different migration patterns. Can this be considered as the observations are discussed?

p. 13, last paragraph and Figure 4C: An added loading control would help the reader appreciate the changes in AMOT levels described in the text. The reduction of PARylation on AMOT, brought about by co-expression of RNF166, is interesting. Is this also observed for tankyrase? Perhaps it is possible to assess this within the same experiment, as tankyrase was co-expressed. It would be interesting to see whether RNF166 simply masks sites that are otherwise targeted for PARylation, or whether tankyrase activity itself is regulated by RNF166.

Figure 6E is not very convincing, likely due to the availability of endogenous ubiquitin, as pointed out by the authors. Could the ubiquitin silencing system be used to corroborate the analysis?

The authors systematically analyse ubiquitylation of tankyrase by various E3 ubiquitin ligases. This is excellent. Amongst others, HUWE1 is identified as a TNKS binder, and there is evidence for ubiquitylation through a mobility shift. For the sake of completeness, the study would benefit from an added experiment to explore whether HUWE1 indeed ubiquitylates tankyrase, as was done for the other candidates.

Whether targets are mono- or di-ubiquitylated cannot always be determined with absolute certainty by

Western blotting. In these cases, can a more careful wording be used to acknowledge this?

Discussion, first paragraph: It is not known whether tankyrase is constitutively active.

Discussion, last paragraph: It remains vague how the study guides the use of tankyrase inhibitors. Some more elaboration would be helpful.

Minor points

The order of panels in the figures is often non-intuitive. Can this be made more reader-friendly?

Where representative experiments are shown, can the authors indicate in the figure legends how many experiments with similar findings were performed in total?

Figure 1: At times, RNF146, 166 and 114 are labelled only by their numbers on the left-hand side of the gel. Can they be labelled "RNFxxx" so the labels are not mistaken for molecular weights?

Some input panels may benefit from loading controls, if these are available. They are often not critical given the rationale of the experiment, but would be a welcome addition where levels are compared.

The section headings may benefit from some rewriting.

A common (but minor) issue: it is cells that are transfected, not constructs.

p. 7: Paragraphs 2 and 3 describe the determination of the regions necessary and sufficient for the RNF166 interaction with tankyrase. The wording chosen by the authors is "important for binding" and "minimal domains required". Can this be changed to "necessary" and "sufficient", respectively, for a more precise description? (In fact, the wording "necessary" and "sufficient" is used subsequently, on page 8.)

p. 8, paragraph 2: "ubiquitin-binding"  "ubiquitin binding"

Figure 2G, last line: "effects [on] TNKS1" or "[a]ffects TNKS1"

p. 8, last paragraph: The tankyrase-binding motifs are potential motifs and, judging by sequence, not particularly strong.

p. 10, 11, 14 (multiple occurrences): "KO"  "K0"

p. 17, paragraph 2: "Since, appearance of this species ..."  "Since appearance of this species ..."

p. 17, paragraph 2: "... candidates for E3s ligases ..."  "... candidates for E3 ligases ..."

Reviewer #2 (Remarks to the Author):

In this manuscript, authors studied regulation of TNKS function and stability by RING-UIM E3 ligase RNF166 and RNF114. Authors performed extensive biochemical studies and their data suggest that RNF166 (and RNF114) bind to mono-ubiquitinated TNKS, promote K11-linked di-ubiquitination, and inhibit RNF146-mediated poly-ubiquitination and degradation of TNKS and TNKS substrate. Overall, this is a nice study with clean data. Finding that TNKS is regulated by another set of E3s is interesting and potentially important. The manuscript is also well written. However, there are several issues that

authors should address. The main issue is that authors heavily relied on overexpression studies, and there is very little evidence that endogenous RNF166 or RNF114 regulates TNKS. Without clean-cut loss-of-function data, it remains possible that the functional data described in the manuscript are overexpression artifacts.

1. The only loss-of-function data is in Fig. 6G. Authors showed that single RNF166 siRNA strongly decreased the protein expression of AMOT. There are three major problems in this figure. First, authors only used one siRNA, so it is completely possible that the activity is off-target. Second, RNF166 siRNA clearly decreased AMOT expression in TNKS DKO cells, although the degree of decrease is smaller in TNKS DKO cells as compared to parental cells. The important caveat is that AMOT is stabilized in TNKS DKO cells, so any interference of transcription and translation of AMOT will show stronger effect on AMOT protein expression in parental cells. The finding that AMOT siRNA decreased AMOT expression in TNKS DKO cells is consistent with the possibility that the effect of siRNA is mediated by some kind of toxicity. Third, RNF166 siRNA clearly increased the protein expression of endogenous TNKS. This seems to be completely against the model proposed by authors. Authors showed previously overexpression of RNF166 stabilized TNKS. It is critical for authors to use two independent CRISPR gRNAs to knockout endogenous RNF166 (or in combination with RNF114) and check the expression of TNKS and AMOT. I would suggest authors to use pool of knockout cells to avoid clonal variation. Other TNKS substrates should also be examined. It would not make much sense if this mechanism would only regulate AMOT.

2. There are alternative explanations of author's key findings. Since RNF166 binds to SAMP-PARP domain of TNKS, it is possible that overexpressed RNF166 binds to the PARP domain and interferes its catalytic activity. Authors examined auto-PARsylation of TNKS using anti-PAR antibody (middle panel of Fig. 1G). Strangely, TNKSi decreased the intensity of top two bands but increased the intensity of the bottom band. Authors should comment on this finding and consider GST-WWE pulldown experiment. The possibility that overexpression of RNF166 inhibits TNKS is also consistent with the finding that overexpression of RNF166 decreased PARsylation of AMOT (Fig. 6G), which authors did not explain. It is also possible that RNF166 itself is a substrate of TNKS, so overexpression of RNF166 redirects the catalytic activity of TNKS from AMOT to RNF166. Unfortunately, it is difficult to rule out this possibility as the RING domain mutant had much weaker binding to TNKS. Related to this, authors did not explain why the RING domain is required for TNKS binding.

3. Overexpression of RNF166 increased the expression of TNKS in the presence of TNKSi (compare lane 2 and lane 4 in Fig. 1G). It also increased the expression of catalytic dead TNKS (compare lane 7 and lane 12). These results are not consistent with authors' model.

Reviewer #3 (Remarks to the Author):

Perrard and Smith present an interesting investigation into novel and non-canonical binding partners of tankyrase. They have shown that RING-UIM E3 ligases bind the SAM-PARP domain instead of the ankyrin repeats of tankyrase, which in itself is an interesting finding. The study primarily focuses on the RING-UIM E3 ligase known as RNF166. The RNF166-TNKS interaction leads to tankyrase stabilization (rather than degradation) via K11-linked diubiquitylation of monoubiquitylated tankyrase. They further identified a canonical binding partner of tankyrase, Angiomotin, that is ubiquitylated by RNF166 and thus protected from TNKS1-mediated degradation through the formation of a ternary complex. They have additionally identified more PAR-binding E3 ligases that target tankyrase for ubiquitylation: CHFR and HUWE1 that lead to TNKS1 degradation similar to RNF146, as well as the DTX family that stabilizes monoubiquitylated TNKS1. The authors have written a compelling paper that significantly advances knowledge in the field of tankyrase biology. The primary conclusions are often shown using more than one approach (e.g. TNKS inhibitors and catalytic mutants, pull-downs with different partners tagged), thus the study is quite thorough and robust. There are a few points that if addressed could potentially improve the clarity and scope of the manuscript; however, the study as it stands is already quite strong and makes important contributions to an active area.

**The authors correctly state that "Tankyrase 1 and 2 have the same binding partners and mostly overlapping functions" and therefore group them together for the remainder of the paper. However, as the phenotypes following knockout in mice are distinct, a mention of the fact that the observed results for TNKS1 would not necessarily translate, or at least not in the same way, to TNKS2 is warranted.

**The study examines interactions using co-immunoprecipitation and mutational analysis. To confirm direct binding, it may be appropriate to look at whether purified TNKS1 SAM-PARP has a robust interaction with full-length or a subdomain construct of a RING-UIM. Given the complexity of the modifications required for interaction, it is understood that this might not be a straightforward experiment.

**Extending from the previous point, the authors state that the SAM-PARP domain is responsible for RING-UIM binding. Have the authors investigated further whether it is specifically the SAM or PARP domain? An investigation into binding of RING-UIMs with either the isolated SAM or PARP domain of TNKS1 might be useful to improve understanding of this novel mode of tankyrase interaction.

**The possibility of a mono-ADP-ribose mark on TNKS1 being responsible for the Di19 interaction is quite interesting. If it is possible to develop this further, for example by monitoring SAM-PARP ADP-ribose modification status, it could add another level of understanding of the system.

** TBS – tankyrase binding site, is never defined in the manuscript

**page 13, "As shown in Fig. 4C"

There were several references to Fig. 4 in this paragraph that I believe should be Fig. 6.

**zinc finger is defined to be ZnF, but ZnFn is mostly used in the text.

**The most up-to-date structural understanding of TNKS would probably indicate 25 ankyrin repeats, but the manuscript states 24 ankyrin repeats, which is probably based on the early bioinformatic analysis of TNKS sequences.

REVIEWER COMMENTS

Reviewer #1 (Remarks to the Author):

This is an exciting study that offers important new insights into the regulation of the stability of tankyrase and its substrate/client proteins in a wide range of biological processes. Through a comprehensive set of experiments building on initial observations by Li et al., 2017, the authors show that two E3 ligases of the RING-UIM family, RNF166 and RNF114, promote the K11-linked di-ubiquitylation of tankyrase and AMOT, a model tankyrase binder. Di-ubiquitylation competes with RNF146-mediated poly-ubiquitylation, thereby giving rise to the stabilisation of tankyrase and a subset of its substrates. The authors also propose a number of E3 ubiquitin ligases responsible for PARylation activity-dependent mono-ubiquitylation of tankyrase and its clients; these mono-ubiquitylated products could serve as substrates for the subsequent modification to generate K11-linked di-ubiquitin. The findings substantially further our understanding of the molecular events downstream of tankyrase-dependent PARylation. Given the diverse links of tankyrase to various (in part disease-relevant) biological processes, the findings are of broad relevance.

The study is comprehensive and well-conducted. The experiments are thorough. Please see below for a number of relatively small suggestions that could further improve an already strong manuscript.

Specific points

p. 3: Referring to TNKS/TNKS2 collectively as tankyrase is helpful, but I suggest to not use the gene symbol "TNKS" to do so (end of penultimate paragraph), as this is reserved for tankyrase 1. Especially as paralogue-specific functions are emerging, clearly distinguishing paralogues where necessary remains important. We agree and now refer to them collectively as "tankyrases".

p. 3, last paragraph: "Colorectal cancer screen" is vague. The Wnt/beta-catenin pathway is described not entirely correctly. AXIN degradation occurs late and is not responsible for the acute activation of the pathway. Instead, the consensus in the field is that the destruction complex is inactivated upon Wnt stimulation, which initiates its remodelling to a Wnt signalosome. In mammalian cells, AXIN1 may not be the concentration-limiting factor of the destruction complex, but APC (PMID: 28810742).

We corrected this. The goal was just to put the role of tankyrase and protein degradation into historical perspective. We changed "colorectal screen" to "a screen for modulators of the Wnt signaling pathway" and now describe Axin as a "scaffolding component". We mention the role of tankyrase and Axin degradation only in the context of the basal state in the absence of Wnt stimulation. We do not go into detail regarding the Wnt signalosome as it is not a focus of the paper.

p. 5, paragraph 2, and subsequent: "TBS" is rather unconventional: can "TBM" be used? Can you also point out that these are "candidate" or "potential" TBMs?

We changed TBS to TBM and indicated "candidate" or "potential" in each case except for TRF1, where it is a validated TBM.

Figure 1A: Can all eight residues of the potential tankyrase binding motifs be spelled out? This way, it will be evident that the motifs themselves are likely weak.

We now spell it out in the Figure legend. We wanted to keep it as is in the Figure so the terminal essential Gs are clearly indicated with their amino acid numbers (19 and 47) since this is how the mutants are named in the Figure.

p. 5, paragraph 2: What is the rationale of using the tankyrase inhibitor at such a high concentration?

The paper describing the inhibitor by Lari Lehtio (Haikarainen et al., ChemMedChem 2013) showed that TNKSi#8 inhibited tankyrase at a concentration of 1 μ M. However, we sometimes found that it did not inhibit

TNKS very well at that concentration (based on stabilization of TNKS). Since the paper showed that at 10 uM TNKSi#8 did not inhibit PARP1 or 2, we felt it was OK to use it at that concentration. We never observed toxic effects of the inhibitor at that concentration.

p. 6, paragraph 2: Referring to Figure 1E, the authors suggest that the stabilising effect of RNF166 and RNF144 on tankyrase is affected by tankyrase inhibition. This does not appear to be the case, at least in Figure 1E.

We were referring to the higher molecular weight forms of TNKS1 and now state this more clearly: “the shift in migration of TNKS1 induced by RNF114 and RNF166 in the Input samples was abrogated by TNKSi treatment (Fig. 1E, lanes 4 and 6), as was the robust immunoprecipitation (Fig. 1E, lanes 14 and 16).”

Also, as requested below, we have now added the amido black panel to the input and IP as loading controls.

p. 6, paragraph 3: "stimulation of tankyrase [levels]" (to to distinguish this from activity)

We added “levels”.

Figure 1G: It is remarkable to obtain such a clear Western blot signal for PAR from a cell lysate. Out of curiosity, do the authors know the identity of the strong single band that appears to accumulate in the anti-PAR blot upon tankyrase inhibition?

This band is PARP1. We provide new data to show that the band is detected by PARP1 antibody and it is abrogated by treatment with the PARPi Olaparib; new data Fig. S1B. Regarding the increase in intensity of the bottom band, it has been observed that when one PARP is inhibited or knocked down, a remaining PARP can show increased PARylation. Hence, the bottom band (PARP1) increases when TNKS is inhibited.

Figure 1H: The authors could quantify additional signals, for example tankyrase levels upon tankyrase inhibition or the extent of PARylation, if feasible.

We provide new data quantifying tankyrase levels (-) and (+) tankyrase inhibition (new data: Fig. 1G) and on the extent of PARylation (-) and (+) tankyrase inhibition (new data: Fig. 1H).

Figure 2D: For the sake of completeness, the schematic of the deltaUIM mutant variant could be added.

The schematic of the delta UIM mutant is shown in Fig. 2A.

p.9, paragraph 1: "This is the first example of a tankyrase binding protein [other than tankyrase itself] that binds to the SAMPARP ...". Recent studies show extensive self-interactions of SAMPARP. The same applies to

p. 18, paragraph 2.

Done.

Figure 3B: What is the explanation for the second band appearing for TRF1, around 40 kDa? It is not observed in panel A.

This is a breakdown product that we frequently see upon overexpression of TRF1. We have now noted that with an asterisk in the figure and legend.

p.9, last paragraph: "Note that we also detect some unmodified TNKS, which is likely due to its ability to oligomerize ..." - The clarity of this statement could be improved by explaining that unmodified tankyrase could potentially heteropolymerise with modified tankyrase in the experiment.

We change the sentence to “which could be due to heteropolymerization of unmodified tankyrase with modified tankyrase.”

Figure 4B: Could the anti-Myc panel be expanded to include higher-molecular-weight species, similar to the equivalent panel in Figure 4C? This will facilitate comparison of the experiments.

Done.

Figure 4C: Co-expression of MycSAMPARP with HA-Ub and either RNF114 or RNF166 gives rise to slower-migrating species likely corresponding to di-ubiquitylated forms of MycSAMPARP. An additional control condition without the expression of any E3 ligase (just MycSAMPARP and HA-Ub) would have been desirable. We have repeated this experiment with this additional control: new data Fig. 4C.

p. 10, paragraph 3: As the authors do not distinguish between mono-ubiquitylation on multiple sites and di-ubiquitylation on a single site, it would be helpful to interpret Figure 4C accordingly at this stage in the manuscript. This would also nicely lead into the subsequent paragraph.

We now make that distinction in the paragraph discussing Fig. 4C.

Figure 4D: Can the anti-Myc input panel be extended towards higher molecular weight to match the corresponding IP panel?

Done.

p. 12, paragraph 3 and Figure 5A: The authors state that expression of RNF166 leads to tankyrase stabilisation. However, this cannot be appreciated from the figure, in either the IP or the input samples. As levels are compared, the input samples require a loading control for analysis.

We mention just the IP samples, where we think there is a clear increase in TNKS1. For the Input, we think there is a discernable increase, but it is weaker. We added the amido black stain, which provides a loading control of crude extract for the Input and an IgG control for the IP.

p. 13, paragraph 1 and Figure 5B: Assessing the abundance of tankyrase is complicated by the different migration patterns. Can this be considered as the observations are discussed?

We have added this consideration to the observations on Fig. 5A, which will then apply to 5B as it is the same format.

p. 13, last paragraph and Figure 4C:

The reviewer is referring to 6C (not 4C); we had mislabeled that - now corrected.

An added loading control would help the reader appreciate the changes in AMOT levels described in the text.

We added the amido black stain, which provides a loading control of crude extract for the Input and an IgG control for the IP.

The reduction of PARylation on AMOT, brought about by co-expression of RNF166, is interesting. Is this also observed for tankyrase? Perhaps it is possible to assess this within the same experiment, as tankyrase was co-expressed.

This would be difficult to do for TNKS. We would have to elute TNKS from the AMOT IP, re-immunoprecipitate it, and then blot for PAR. We think this would be a very challenging experiment.

It would be interesting to see whether RNF166 simply masks sites that are otherwise targeted for PARylation, or whether tankyrase activity itself is regulated by RNF166.

Yes, we agree this would be very interesting. We showed that RNF166 binds to the SAMPARP (and not the ankyrin domain) of TNKS. Such binding offers an opportunity for formation of a ternary complex between AMOT, TNKS and RNF166. As we observed that PARylation of AMOT was reduced, RNF166 binding to SAMPARP may limit the ability of TNKS to PARylate AMOT. Future experiments using purified components should allow us to test this hypothesis.

Figure 6E is not very convincing, likely due to the availability of endogenous ubiquitin, as pointed out by the authors. Could the ubiquitin silencing system be used to corroborate the analysis?

We repeated this analysis using the ubiquitin silencing system. It is more convincing, particularly for the K11 versus the K11R mutant - we see clear diubiquitylation versus monoubiquitylation: new data. Fig. S3A.

The authors systematically analyse ubiquitylation of tankyrase by various E3 ubiquitin ligases. This is excellent. Amongst others, HUWE1 is identified as a TNKS binder, and there is evidence for ubiquitylation through a

mobility shift. For the sake of completeness, the study would benefit from an added experiment to explore whether HUWE1 indeed ubiquitylates tankyrase, as was done for the other candidates.

We provide new data to show that HUWE1 stimulates ubiquitylation of TNKS: new data Fig. 7I.

Whether targets are mono- or di-ubiquitylated cannot always be determined with absolute certainty by Western blotting. In these cases, can a more careful wording be used to acknowledge this?

We added “Although western blotting does not permit a precise determination of size, their migration is consistent with” on page 11 (for Fig. 4C) and on page 15 (for Fig. 6C and 6F)

Discussion, first paragraph: It is not known whether tankyrase is constitutively active.

We removed it.

Discussion, last paragraph: It remains vague how the study guides the use of tankyrase inhibitors. Some more elaboration would be helpful.

We changed “new uses for tankyrase inhibitors in cancer therapy” to “new strategies targeting tankyrases in human disease.”, which relates to the previous sentence “..... may promote new protein-protein interactions and reroute TNKS1 to other pathways including autophagy, innate immunity, and cell cycle regulation.”

Minor points

The order of panels in the figures is often non-intuitive. Can this be made more reader-friendly?

We have moved some panels to the supplemental data to make things more clear.

Where representative experiments are shown, can the authors indicate in the figure legends how many experiments with similar findings were performed in total?

Done.

Figure 1: At times, RNF146, 166 and 114 are labelled only by their numbers on the left-hand side of the gel. Can they be labelled "RNFxxx" so the labels are not mistaken for molecular weights?

Done.

Some input panels may benefit from loading controls, if these are available. They are often not critical given the rationale of the experiment, but would be a welcome addition where levels are compared.

We have added amido black panels as loading controls for the Input and IPs in Fig. 1E, 5A, 6C, 7G, 7I, and S2A.

The section headings may benefit from some rewriting.

OK.

A common (but minor) issue: it is cells that are transfected, not constructs.

We see it written either way in the literature.

p. 7: Paragraphs 2 and 3 describe the determination of the regions necessary and sufficient for the RNF166 interaction with tankyrase. The wording chosen by the authors is "important for binding" and "minimal domains required". Can this be changed to "necessary" and "sufficient", respectively, for a more precise description? (In fact, the wording "necessary" and "sufficient" is used subsequently, on page 8.)

Done.

p. 8, paragraph 2: "ubiquitin-binding"  "ubiquitin binding"

Done.

Figure 2G, last line: "effects [on] TNKS1" or "[a]ffects TNKS1"

Done.

p. 8, last paragraph: *The tankyrase-binding motifs are potential motifs and, judging by sequence, not particularly strong.*

We added "potential".

p. 10, 11, 14 (multiple occurrences): "KO"  "K0"

Done.

p. 17, paragraph 2: "Since, appearance of this species ..."  "Since appearance of this species ..."

Done.

p. 17, paragraph 2: "... candidates for E3s ligases ..."  "... candidates for E3 ligases ..."

Done.

Reviewer #2 (Remarks to the Author):

In this manuscript, authors studied regulation of TNKS function and stability by RING-UIM E3 ligase RNF166 and RNF114. Authors performed extensive biochemical studies and their data suggest that RNF166 (and RNF114) bind to mono-ubiquitinated TNKS, promote K11-linked di-ubiquitination, and inhibit RNF146-mediated poly-ubiquitination and degradation of TNKS and TNKS substrate. Overall, this is a nice study with clean data. Finding that TNKS is regulated by another set of E3s is interesting and potentially important. The manuscript is also well written. However, there are several issues that authors should address. The main issue is that authors heavily relied on overexpression studies, and there is very little evidence that endogenous RNF166 or RNF114 regulates TNKS. Without clean-cut loss-of-function data, it remains possible that the functional data described in the manuscript are overexpression artifacts.

Yes, the paper does rely on overexpression studies. Analyses of post-translational modifications like ubiquitylation and PARylation are challenging. In keeping with state-of-the-art techniques in the literature, this type of analysis often relies upon overexpression. This manuscript reports multiple novel E3 ligases that interact with tankyrase and identifies the nature of the interactions and specifics of the modifications. We feel it is unlikely that this body of work reflects overexpression artifacts. Unfortunately, due to the complexity of the multiprotein interactions and modifications, particularly regarding the balance of degradation and stabilization, it is not straightforward to provide clean-cut loss of function data on endogenous proteins at this time. Nonetheless, we hope the reviewer can see that the work contributes greatly to the field and lays the groundwork for future functional analyses.

1. The only loss-of-function data is in Fig. 6G. Authors showed that single RNF166 siRNA strongly decreased the protein expression of AMOT. There are three major problems in this figure.

We are grateful to the reviewer for their careful analysis of Fig. 6G and address their concerns below.

First, authors only used one siRNA, so it is completely possible that the activity is off-target.

The reviewer is correct. We tried additional siRNA and shRNA and did not reproduce the result – details below.

Second, RNF166 siRNA clearly decreased AMOT expression in TNKS DKO cells, although the degree of decrease is smaller in TNKS DKO cells as compared to parental cells. The important caveat is that AMOT is stabilized in TNKS DKO cells, so any interference of transcription and translation of AMOT will show stronger effect on AMOT protein expression in parental cells. The finding that AMOT siRNA (we think the reviewer here refers to RNF166 siRNA) decreased AMOT expression in TNKS DKO cells is consistent with the possibility that the effect of siRNA is mediated by some kind of toxicity.

We did not see any signs of toxicity, but the reviewer's point about AMOT levels being higher in the DKO cells (and therefore showing a weaker effect) is well taken.

Third, RNF166 siRNA clearly increased the protein expression of endogenous TNKS.

This is an off-target effect as the reviewer suggested. With additional RNAi experiments we do not see a change in TNKS levels – details below.

This seems to be completely against the model proposed by authors. Authors showed previously overexpression of RNF166 stabilized TNKS.

We did not propose a simple model where depletion of RNF166 and/or 114 would lead to a reduction in TNKS. Mechanistically, the regulation relies on competition and formation of ternary complexes and thus the level of each of the players may be critical. This may not be easily recapitulated in the cell lines we use. For example, the levels of endogenous RNF166 are very low in these cell lines and thus they may not be engaged in regulation of TNKS expression levels. The goal of future studies will be to determine how these interactions are triggered. For example, future studies will investigate examples where associations between tankyrases, the RNF family members, and signaling pathways are triggered by the innate immunity (discussed in more detail in the Results page 13 and Discussion pages 19 and 20.)

It is critical for authors to use two independent CRISPR gRNAs to knockout endogenous RNF166 (or in combination with RNF114) and check the expression of TNKS and AMOT. I would suggest authors to use pool of knockout cells to avoid clonal variation.

To address the reviewer's concern, we generated new reagents for knockdown. We used a second distinct siRNA and an inducible lentiviral shRNA to deplete RNF166. We observed efficient depletion of RNF166, but did not recapitulate the result with the first siRNA. We did not detect an increase (or any change) in TNKS, nor did we see a reduction in AMOT. Thus, we agree with the reviewer's interpretation that the result in Fig. 6G (despite being reproducible and specific) is likely an off-target effect. We have deleted panel G from Fig. 6 and again, we are grateful to the reviewer for detecting this.

We have now performed analyses with multiple siRNAs and shRNAs for RNF166 and/or RNF114 and we do not see a reduction in TNKS levels. As described above, we did not anticipate that depletion of 166/114 would lead to a reduction in TNKS levels in these cells. Going forward, once we establish conditions for measuring formation of endogenous ternary complexes, we can then use these reagents to evaluate the role of RNF166 and RNF114.

Other TNKS substrates should also be examined. It would not make much sense if this mechanism would only regulate AMOT.

We provide new data where we performed HA-ubiquitin immunoprecipitation and show that RNF166 plus TNKS stimulate diubiquitylation of AMOT (endogenous). We then used this format to probe the status of the tankyrase binding protein TAB182 (endogenous) (which we showed in Supplemental Table 1 was highly enriched in the TNKS1/RNF166 immunoprecipitate) and show increased ubiquitylation of TAB182 dependent on RNF166 and TNKS1: new data Fig. 6E.

2. There are alternative explanations of author's key findings. Since RNF166 binds to SAMP-PARP domain of TNKS, it is possible that overexpressed RNF166 binds to the PARP domain and interferes its catalytic activity. Authors examined auto-PARylation of TNKS using anti-PAR antibody (middle panel of Fig. 1G). Strangely, TNKSi decreased the intensity of top two bands but increased the intensity of the bottom band. Authors should comment on this finding and consider GST-WWE pulldown experiment.

We provide new data showing that the top two bands in the anti-PAR blot are due to TNKS; they are inhibited by TNKSi and not by the PARP1 inhibitor olaparib, whereas the bottom band is due to PARP1; it is inhibited by the PARP1 inhibitor Olaparib and not by TNKSi and is detected in an anti-PARP1 immunoblot: new data Fig. S1B. Regarding the increase in intensity of the bottom band, it has been observed that when one PARP is inhibited or knocked down, a remaining PARP can show increased PARylation. Hence, the bottom band (PARP1) increases when TNKS is inhibited.

We provide new data for the middle panel of Fig. 1G (now Fig. 1F) quantifying the effect of RNF166 and RNF114 on TNKS1 PARylation and show that they stimulate PARTNKS1 levels (new data Fig. 1H). Thus, they do not inhibit PARTNKS1 levels, indicating that the catalytic activity of TNKS is not reduced by RNF166 or RNF114.

The possibility that overexpression of RNF166 inhibits TNKS is also consistent with the finding that overexpression of RNF166 decreased PARylation of AMOT (Fig. 6G) which authors did not explain.

We believe the reviewer is referring to Fig. 6C (not 6G) as it is the only panel with PARylated AMOT.

We did explain the result. We said in the Results section that we observed a "reduction in AMOT PARylation (lane 10, second panel). These data suggest that RNF166-mediated ubiquitylation of AMOT protects it from PARylation by TNKS1 and subsequent degradation." And further, we proposed a mechanism in the Discussion: "We show that RNF166 binds to the SAMPARP domain of TNKS. This is the first example of a TBP that binds SAMPARP and not the ankyrin domain. Such binding offers an opportunity for formation of a ternary complex, as shown for AMOT, TNKS and RNF166. We showed that in this setting PARylation of AMOT was reduced. RNF166 binding to SAMPARP may limit the ability of TNKS to PARylate a bound TBP, which would limit its

interaction with RNF146. At the same time RNF166 (by binding and capping the monoUb and by K11 diubiquitylation) could block elongation of ubiquitin chains. Overall, this could provide an effective counter to RNF146-mediated degradation of AMOT, resulting in AMOT stabilization”

It is also possible that RNF166 itself is a substrate of TNKS, so overexpression of RNF166 redirects the catalytic activity of TNKS from AMOT to RNF166. Unfortunately, it is difficult to rule out this possibility as the RING domain mutant had much weaker binding to TNKS.

We provide new data to show that RNF166 is not a substrate for TNKS. We performed immunoprecipitation of RNF146 (which is a target of TNKS) and RNF166 side-by-side and show that RNF146 is PARylated whereas RNF166 (which is present at the same level as RNF146) is not PARylated: new data Fig. S1D.

Related to this, authors did not explain why the RING domain is required for TNKS binding.

We showed that the RING domain is actually not required for binding. The Di19 domain is necessary and sufficient, the UIM stimulates it, and the RING-C2HC further stimulates it (Fig. 2). We think the RING-C2HC domain may be important for the structural integrity of the whole protein and the ubiquitylation could further impact binding.

3. Overexpression of RNF166 increased the expression of TNKS in the presence of TNKSi (compare lane 2 and lane 4 in Fig. 1G).

Yes, RNF166 leads to an increase in TNKS level even in the presence of TNKSi. We now point this out on page 6 and state: “We observed some increase in TNKS1 in the presence of TNKSi, but only with RNF166 (not RNF114) and only on the unmodified form of TNKS1”

It also increased the expression of catalytic dead TNKS (compare lane 7 and lane 12).

Yes. We now point this out on page 6 “The level of TNKS1 CD in the Input was stimulated by RNF166, but again (as described above in Fig. 1F for TNKSi), the effect was only with RNF166 and only on the unmodified form of TNKS1. This appears to be a distinct effect of RNF166 that will not be pursued further here.”

These results are not consistent with authors’ model.

These results do not relate to our model since the effect is independent of TNKS catalytic activity, does not induce modification of TNKS1, and only applies to RNF166 (not RNF114). While this effect could certainly be interesting and a topic for future study, we think it does not directly impact our results.

Reviewer #3 (Remarks to the Author):

Perrard and Smith present an interesting investigation into novel and non-canonical binding partners of tankyrase. They have shown that RING-UIM E3 ligases bind the SAM-PARP domain instead of the ankyrin repeats of tankyrase, which in itself is an interesting finding. The study primarily focuses on the RING-UIM E3 ligase known as RNF166. The RNF166-TNKS interaction leads to tankyrase stabilization (rather than degradation) via K11-linked diubiquitylation of monoubiquitylated tankyrase. They further identified a canonical binding partner of tankyrase, Angiomotin, that is ubiquitylated by RNF166 and thus protected from TNKS1-mediated degradation through the formation of a ternary complex. They have additionally identified more PAR-binding E3 ligases that target tankyrase for ubiquitylation: CHFR and HUWE1 that lead to TNKS1 degradation similar to RNF146, as well as the DTX family that stabilizes monoubiquitylated TNKS1. The authors have written a compelling paper that significantly advances knowledge in the field of tankyrase biology. The primary conclusions are often shown using more than one approach (e.g. TNKS inhibitors and catalytic mutants, pull-downs with different partners tagged), thus the study is quite thorough and robust. There are a few points that if addressed could potentially improve the clarity and scope of the manuscript; however, the study as it stands is already quite strong and makes important contributions to an active area.

***The authors correctly state that “Tankyrase 1 and 2 have the same binding partners and mostly overlapping functions” and therefore group them together for the remainder of the paper. However, as the phenotypes following knockout in mice are distinct, a mention of the fact that the observed results for TNKS1 would not necessarily translate, or at least not in the same way, to TNKS2 is warranted.*

We deleted the statement about grouping them together for the remainder of the paper. We now specifically refer to TNKS1 throughout the paper since that is what we analyze in all the experiments. In the discussion we use the term tankyrases a few times but that is in a very broad sense.

***The study examines interactions using co-immunoprecipitation and mutational analysis. To confirm direct binding, it may be appropriate to look at whether purified TNKS1 SAM-PARP has a robust interaction with full-length or a subdomain construct of a RING-UIM. Given the complexity of the modifications required for interaction, it is understood that this might not be a straightforward experiment.*

In order to have a robust interaction the SAMPARP domain would have to be monoubiquitylated. This would require that we perform the ubiquitylation of SAMPARP in vitro with a PAR-binding E3 ligase and then perform a binding assay with purified RING-UIM. We would like to do these experiments, but they may take some time and we feel it is outside the scope of the paper.

***Extending from the previous point, the authors state that the SAM-PARP domain is responsible for RING-UIM binding. Have the authors investigated further whether it is specifically the SAM or PARP domain? An investigation into binding of RING-UIMs with either the isolated SAM or PARP domain of TNKS1 might be useful to improve understanding of this novel mode of tankyrase interaction.*

We agree that it would be useful information and tried immunoprecipitation analysis with the individually expressed domains. Unfortunately, the results were not informative. We performed immunoprecipitation with RNF166 and tested each domain for co-IP. For the PARP domain, we found that it did not co-IP with RNF166. The problem here is that PARP domain alone is not active -it requires the SAM domain, thus it may not be in a physiological formation. For the SAM domain, we found that it did co-IP with RNF166, but it was non-specific (it coimmunoprecipitated with the control BAP as well). The problem here is that the SAM domain by itself multimerizes extensively and thus may precipitate non-specifically. So, unfortunately the experiments were inconclusive.

***The possibility of a mono-ADP-ribose mark on TNKS1 being responsible for the Di19 interaction is quite interesting. If it is possible to develop this further, for example by monitoring SAM-PARP ADP-ribose modification status, it could add another level of understanding of the system.*

The reviewer makes a great point. We provide new data showing that full-length TNKS and the SAMPARP domain are MARYlated: new data Fig. S2. We think this really brings the story to a new level and opens up a new area of investigation for the future now discussed in the Discussion on pages 18 and 19. We thank the reviewer for a great question!

*** TBS – tankyrase binding site, is never defined in the manuscript*

We changed it to TBM and define it in the first paragraph of the Introduction.

***page 13, "As shown in Fig. 4C"*

There were several references to Fig. 4 in this paragraph that I believe should be Fig. 6.

We corrected them.

***zinc finger is defined to be ZnF, but ZnFn is mostly used in the text.*

We corrected them to ZnF.

***The most up-to-date structural understanding of TNKS would probably indicate 25 ankyrin repeats, but the manuscript states 24 ankyrin repeats, which is probably based on the early bioinformatic analysis of TNKS sequences.*

We remove the 24 and just refer to it as “an ankyrin repeat domain”.

REVIEWERS' COMMENTS

Reviewer #1 (Remarks to the Author):

I thank the authors for the careful revisions, which sufficiently address all the points I had raised. This is a very interesting study, and I look forward to seeing it published.

As a minor point, it would still help to spell out that the same high concentration of TNKSi#8 was chosen previously, and that under those conditions TNKSi#8 did not inhibit PARP1/2.

Reviewer #2 (Remarks to the Author):

Authors have addressed many of my concerns. I recognize importance of the work and complexity of the system. Nevertheless, it would be a stronger paper if loss of function data can be provided.

Reviewer #3 (Remarks to the Author):

The authors have addressed the concerns/suggestions that we raised in the initial review. The addition of the analysis of TNKS mono-ADP-ribose modification is nice. Some of the other experiments did not quite pan out due to technical considerations, but I do not think that the lack of these other experiments limits or impacts the key conclusions of the study.

REVIEWERS' COMMENTS

Reviewer #1 (Remarks to the Author):

I thank the authors for the careful revisions, which sufficiently address all the points I had raised. This is a very interesting study, and I look forward to seeing it published.

As a minor point, it would still help to spell out that the same high concentration of TNKSi#8 was chosen previously, and that under those conditions TNKSi#8 did not inhibit PARP1/2.

We now state on page 20 that TNKSi was used "at a concentration (10 μ M) that does not inhibit PARP1 or 2³⁶."

Reviewer #2 (Remarks to the Author):

Authors have addressed many of my concerns. I recognize importance of the work and complexity of the system. Nevertheless, it would be a stronger paper if loss of function data can be provided.

We agree that it will be important going forward to obtain loss of function data. However, at this point in time we feel that it is beyond the scope of the manuscript.

Reviewer #3 (Remarks to the Author):

The authors have addressed the concerns/suggestions that we raised in the initial review. The addition of the analysis of TNKS mono-ADP-ribose modification is nice. Some of the other experiments did not quite pan out due to technical considerations, but I do not think that the lack of these other experiments limits or impacts the key conclusions of the study.

Thank you.